# Development of methodology to support molecular endotype discovery from synovial fluid of individuals with knee osteoarthritis: The STEpUP OA consortium

Yun Deng[1‡], Thomas A. Perry[1‡], Philippa Hulley[2], Rose A. Maciewicz[1],
Joanna Mitchelmore[3], Darryl Perry[4], Staffan Larsson[5], Sophie Brachat[3],
André Struglics[5], C. Thomas Appleton[6], Stefan Kluzek[2,7], Nigel K. Arden[2,8],
David Felson[9], Brian Marsden[2,10], Brian D. M. Tom[11], Laura Bondi[11], Mohit Kapoor[12],
Vicky Batchelor[1], Jennifer Mackay-Alderson[1], Vinod Kumar[2], L. Stefan Lohmander[5], Tim
J. Welting[13], David A. Walsh[14,15], Ana M. Valdes[14], the STEpUP OA Consortium[¶], Tonia
L. Vincent[1☉]*, Fiona E. Watt[1,16☉], Luke Jostins-Dean[1☉]

1 Centre for Osteoarthritis Pathogenesis Versus Arthritis, Kennedy Institute of Rheumatology, NDORMS, University of Oxford, Oxford, United Kingdom, 2 Nuffield Department of Orthopaedics, Rheumatology, and Musculoskeletal Sciences, University of Oxford, Oxford, United Kingdom, 3 Novartis Institutes for Biomedical Research, Basel, Switzerland, 4 SomaLogic, Boulder, Colorado, United States of America, 5 Department of Clinical Sciences Lund, Orthopaedics, Faculty of Medicine, Lund University, Lund, Sweden, 6 Bone and Joint Institute, University of Western Ontario, London, Ontario, Canada, 7 NIHR Nottingham Biomedical Research Centre and Versus Arthritis Sport, Exercise and Osteoarthritis Centre, University of Nottingham, Nottingham, United Kingdom, 8 Centre for Sport, Exercise and Osteoarthritis Research Versus Arthritis, University of Oxford, Oxford, United Kingdom, 9 Section of Rheumatology, Boston University School of Medicine, Boston, Massachusetts, United States of America, 10 Nuffield Department of Medicine, University of Oxford, Oxford, United Kingdom, 11 MRC Biostatistics Unit, University of Cambridge, Cambridge, United Kingdom, 12 Schroeder Arthritis Institute, University Health Network, Toronto, Ontario, Canada, 13 Laboratory for Experimental Orthopedics, Department of Orthopedic Surgery, Maastricht University, Maastricht, Netherlands, 14 Pain Centre Versus Arthritis, Advanced Pain Discovery Platform, and the NIHR Nottingham Biomedical Research Centre, University of Nottingham, Nottingham, United Kingdom, 15 Sherwood Forest Hospitals NHS Foundation Trust, Sutton in Ashfield, United Kingdom, 16 Department of Immunology and Inflammation, Imperial College London, London, United Kingdom

☉ These authors contributed equally to this work.
‡ YD and TAP are joint first authors on this work.
¶ Membership of the STEpUP OA Consortium is provided in the *STEpUP OA Consortium author block* section of the manuscript.
* tonia.vincent@kennedy.ox.ac.uk

**Data Availability Statement:** The minimal datasets upon which this data relies and all R code, including the html vignette, are available at https://

## Abstract

### Objectives

To develop a protocol for largescale analysis of synovial fluid proteins, for the identification of biological networks associated with subtypes of osteoarthritis.

### Methods

Synovial Fluid To detect molecular Endotypes by Unbiased Proteomics in Osteoarthritis (STEpUP OA) is an international consortium utilising clinical data (capturing pain, radiographic severity and demographic features) and knee synovial fluid from 17 participating

github.com/dengyun-git/STEpUp_QC_Paper. The full STEpUP OA dataset may be made available by application to the Data Access and Publication Group of STEpUP OA (stepupoa@kennedy.ox.ac.uk) once the primary analysis manuscript is published, in accordance with what is stipulated in the authors' Consortium Agreement. Neither the minimal dataset nor the full STEpUP OA dataset include patient identifiable data. This may attract a fee for access to cover administrative processing.

**Funding:** The study was supported by Kennedy Trust for Rheumatology Research (grant number: 171806), Versus Arthritis (grant number: 22473), Centre for OA Pathogenesis Versus Arthritis (grant numbers: 21621, 20205), Galapagos, Biosplice, Novartis, Fidia, UCB, Pfizer (non consortium member) and Somalogic (in kind contributions). The funders Kennedy Trust for Rheumatology Research, Versus Arthritis and Pfizer had no role in the study design, data collection and analysis, decision to publish or preparation of the manuscript.The funders Galapagos, Biosplice, Novartis, Fidia, UCB and SomaLogic were all active consortium members, attending consortium meetings. As such they made contributions to the study design and support of data collection, decision to publish and review and commenting on the manuscript. In addition, SomaLogic, UCB and Novartis were members of the Data Analysis Group".

**Competing interests:** YD, TAP, PH, SL, AS, NKA, DF, BM, AMV, SK, VB, JMA and VK declare no conflicts of interest. FW has received consultancy fees from Pfizer, and has a leadership role at the Medical Research Council (panel member) and Osteoarthritis and Cartilage (Associate Editor). LSL has received consultancy fees from Arthro Therapeutics AB, and is an advisory board member of AstraZeneca (non consortium member). LJD has received consultancy fees from Nightingale Health PLC. TLV has no conflicts to declare with the exception of grant income for STEpUP OA from industry partners (see above). RAM is a shareholder of AstraZeneca. SB and JM are employees and shareholders of Novartis (consortium members). MK has received support for attending the Gordon Research Conference, OARSI meeting, International Cartilage Repair Society, Munster University, is a board member of the Dutch Arthritis Society (Chair of Visitation Board), and has a leadership role at Osteoarthritis Research Society International (Board of Directors Member). CTA has received consultancy fees from Novartis, and has received honoraria for educational purposes also from Novartis. DAW has received consultancy fees from GlaxoSmithKline

cohorts. 1746 samples from 1650 individuals comprising OA, joint injury, healthy and inflammatory arthritis controls, divided into discovery (n = 1045) and replication (n = 701) datasets, were analysed by SomaScan Discovery Plex V4.1 (>7000 SOMAmers/proteins). An optimised approach to standardisation was developed. Technical confounders and batch-effects were identified and adjusted for. Poorly performing SOMAmers and samples were excluded. Variance in the data was determined by principal component (PC) analysis.

### Results

A synovial fluid standardised protocol was optimised that had good reliability (<20% co-efficient of variation for >80% of SOMAmers in pooled samples) and overall good correlation with immunoassay. 1720 samples and >6290 SOMAmers met inclusion criteria. 48% of data variance (PC1) was strongly correlated with individual SOMAmer signal intensities, particularly with low abundance proteins (median correlation coefficient 0.70), and was enriched for nuclear and non-secreted proteins. We concluded that this component was predominantly intracellular proteins, and could be adjusted for using an 'intracellular protein score' (IPS). PC2 (7% variance) was attributable to processing batch and was batch-corrected by ComBat. Lesser effects were attributed to other technical confounders. Data visualisation revealed clustering of injury and OA cases in overlapping but distinguishable areas of high-dimensional proteomic space.

### Conclusions

We have developed a robust method for analysing synovial fluid protein, creating a molecular and clinical dataset of unprecedented scale to explore potential patient subtypes and the molecular pathogenesis of OA. Such methodology underpins the development of new approaches to tackle this disease which remains a huge societal challenge.

## Introduction

Osteoarthritis (OA) is a highly prevalent and disabling condition and represents a huge unmet clinical need within the musculoskeletal field [1, 2], now recognised by the U.S. Food and Drug Administration as a 'serious disease' [3, 4]. OA affects synovial joints manifesting as localised, low-grade inflammation of the synovium, cartilage damage and subchondral bone remodeling [5], which lead to pain, stiffness and loss of function [6, 7]. Despite growing clinical demand and best efforts in pre-clinical models and translational studies to understand the underlying pathogenesis, target discovery and drug development for knee OA in humans have been slow [8] and many clinical trials have failed [9–12].

Development of new therapies in OA has been hampered by a lack of detailed understanding of the pathological processes in the joint, and an appreciation of whether or not there are distinct patient subtypes. Our understanding of disease pathology is likely to be advanced through large-scale proteomics which will identify proteins and protein networks associated with OA pathology. Synovial fluid (SF), acquired by joint aspiration, has several advantages over blood for exploring molecular mechanisms. Firstly, it has adjacency to joint tissues, and may reflect activities in synovium, bone as well as cartilage [13–15]. Secondly, concentrations of analytes within the SF provide an indication of biological activity and target tissue activation [16, 17]. Thirdly, the SF from a given joint is less confounded by disease at other sites than is,

plc, AKL Research & Development Limited, Pfizer Ltd, Eli Lilly and Company, Contura International, and AbbVie Inc, has received honoraria for educational purposes from Pfizer Ltd and AbbVie Inc, is a board member of UKRI (Director) and Versus Arthritis Advanced Pain Discovery Platform. "This does not alter our adherence to PLOS ONE policies on sharing data and materials (see also Data Availability statement)".

for example, blood. Finally, a number of analytes that are highly regulated in the SF are not reflected in the plasma [13, 18, 19]. Proteomic profiling of human SF to date has been in low numbers of patient samples, either by examining small numbers of pre-defined analytes or by mass spectrometry (typically identifying around 500 of the most abundant proteins) [20–22]. In order to unlock the full potential of SF, we must first optimise its use and overcome analysis challenges which include high viscosity due to rich hyaluronan content, variability in joint effusion volume between and within patients, and potential contamination with blood at time of aspiration. To combat some of these challenges, hyaluronidase treatment of the fluid post aspiration or lavage of the joint prior to aspiration have been utilised [23]. New high through-put methodology for protein assessment are now available [24, 25] and have shown utility in other conditions including cardiovascular disease [26] and dementia [27]. Such technologies have been predominately used for plasma or serum, which may be particularly relevant to systemic diseases. They have also been employed recently to verify the functional effects of genetic risk variants in disease, including OA [28]. There is a need to develop similar methodology for SF which could then also be examined in the context of genetic risk to unravel key pathways in disease.

OA might not be a single disease [29, 30] but rather a group of diseases with a similar clinical presentation but driven by distinct molecular pathways known as 'endotypes'. These might determine the course of disease and in some cases predict response to treatment. It is presumed that molecular endotypes might relate to discernible patient characteristics and may help to explain the heterogeneity of OA 'clinical phenotypes' [31, 32]. Many cellular processes have been proposed as critical drivers in OA pathogenesis such as immune-mediated inflammation [33], mechanically-mediated inflammation ('mechanoflammation') [34], low/failed tissue repair [8, 35] and cellular senescence [36]. These in turn may relate to a broad range of aetiological factors that are associated with OA [37–41]. Efforts have been made to classify subgroups of people with OA based on epidemiological factors [42], with several clinically defined phenotypes now suggested in the literature [43–46]. A recent systematic review of 24 studies reported that up to 84% of people with OA could be assigned to at least one of six phenotypes [43, 47]. These clinical phenotypes are, however, not mutually exclusive, and are poor systematic classifiers because they are a mixture of overlapping demographic, clinical, radiographic, aetiological and systemic features. They currently have limited clinical applicability [29] and there is a paucity of data relating them to distinct molecular pathways or to clinical outcomes in OA [48].

Whilst there have been a plethora of studies of candidate molecules trying to identify diagnostic or prognostic biomarkers of OA, relatively few have used hypothesis-free approaches in large numbers of human biological samples to identify molecular endotypes [43, 49, 50]. Such collaborations are required to help move biomarker discovery forward [51]. The Synovial fluid To detect molecular Endotypes by Unbiased Proteomics in OA (STEpUP OA) Consortium was set up to address a primary objective: to determine whether there are detectable distinct molecular endotypes in knee OA, through a hypothesis-free, unsupervised proteomic analysis applying SomaLogic array technology [52] of SF from a large number of participants with, or at increased risk of, knee OA. SomaScan, an aptamer-based proteomics technology, offers the ability to measure large numbers of protein analytes from a small volume of biological fluid.

Detailed methodology is lacking for quality control (QC) and data analysis pipelines specifically tailored to SF. In this study we describe the processing and analysis of SF, and the optimisation of a standardised quality control (QC) and analysis pipeline for these data. We evaluate performance of SF on the SomaLogic platform at scale for the first time and identify important technical confounders requiring adjustment prior to downstream analysis. Prespecified potential confounding factors included those relating to sample processing or to the sample itself,

such as its age, number of freeze-thaws, visible blood staining and sample volume. These investigations were used to inform STEpUP OA's primary data analysis plan (https://www.kennedy.ox.ac.uk/oacentre/stepup-oa/stepup-oa) and make our work replicable by others.

## Methods

Details of consortium structure, governance and ethical approvals can be found in S1 File. Working groups oversaw key activities (S1 Fig). Six participating sites with 17 participant collections (henceforth referred to as 'cohorts') including those with either knee OA or acute knee joint injury provided associated SF samples. Each had ethical approval (S1 Table). In addition, the University of Oxford Medical Sciences Central University Research Ethics Committee (CUREC) granted ethical approval for the processing, storage and use of samples and linked data for this project on 1[st] November 2019 (R67029/RE001).

### Participant eligibility criteria

All but one cohort had existing associated stored participant SF samples. Inclusion criteria were: i) evidence of a confirmed diagnosis of knee OA, or history of recent knee injury, ii) associated basic clinical information including (as a minimum) age at sampling, sex and indication of OA disease status, iii) a minimum volume of SF (90 μl, ideally 200 μl) and iv) SF had been centrifuged between 1800-3000g, prior to supernatant storage at -80˚C. Exclusion criteria were: i) additional forms of arthritis e.g. gout, rheumatoid arthritis, psoriatic arthritis, as determined by host investigator; ii) confounding medical conditions e.g. concurrent infection, cancer; iii) confounding treatments e.g. index knee surgery in the preceding 6 months, index knee steroid injection in preceding 3 months; iv) chemotherapy and; v) significant deviation in storage procedure (e.g. freezer drop-out defined by host investigator).

### Sample processing and SomaLogic assay

Consortium samples: 1746 SF samples were eligible and processed for STEpUP OA (S1 File). Archived samples were shipped to Oxford over a 20-month period (received: 27[th] November 2019 to 26[th] August 2021). No authors had access to participant identifiable information. A STEpUP participant ID number (PIN) and related unique sample identification number (SIN) were generated for each participant and their associated sample(s). Sample processing was performed in Oxford in four tranches over a 24-month period. For analysis by SomaLogic, SF enzymatic digestion, using hyaluronidase, was carried out. Briefly, sufficient bovine testicular hyaluronidase (4mg/ml; Sigma-Aldrich) for the entire project was reconstituted as a single batch from a single lot number and frozen in aliquots until use. For each tranche, batch processing was performed over consecutive working days, i.e. over as short a time as possible. Briefly, a batch of SFs was thawed, centrifuged at 3000g at 20˚C for 25 minutes and 175 μl of SF supernatant diluted 1:2 with the same volume of hyaluronidase solution and agitated at room temperature for 1 hour, followed by further centrifugation for 5 minutes [19]. Supernatants were aliquoted and stored at -80˚C and transferred on dry ice by temperature-controlled shipping to SomaLogic (single shipment per tranche).

Consortium controls/QC samples: Equal volumes of SF samples from 6 participants per group were used to generate single batches of hyaluronidase-treated 'pooled samples' for each of OA and knee injury at the start of project. Subaliquots of these then acted as internal QC controls, being run on each SomaLogic plate, enabling calculation of intra-assay and inter-assay coefficients of variation (CVs), as well as assessing effects of freeze thaw (a multiple freeze-thawed aliquot), frozen storage of hyaluronidase (an untreated aliquot freshly treated with frozen hyaluronidase during processing of each tranche) and centrifugation (an additional unspun pooled sample, from 6 unspun OA SF samples). A further 18 samples (split at the

time of collection, with the paired aliquot remaining 'unspun') were included to further examine the effects of centrifugation. 42 other 'comparator' samples were included (disease-free controls from non-painful knees or from normal joints at amputation/post-mortem; samples from individuals with definite inflammatory arthritis). Three samples from three separate participants were re-processed under 3 different temperature conditions and re-analysed to examine the effects of laboratory re-processing. A subgroup of the freshly collected samples were processed specifically to test generalizability to OA SF which had not been centrifuged ('unspun') (n = 18). 235 unspun OA samples were subsequently included in the replication analysis.

Samples were assayed in three 'dilution bins' (to allow accurate quantification of high and low abundant proteins simultaneously), on the SomaLogic SomaScan Discovery Plex V4.1 by Soma-Logic, in Boulder, US. All samples from all 4 tranches were processed as a single batch on twenty-two sequential 96-well plates in January 2022. All samples were randomised within and between plates whilst ensuring appropriate controls on each plate. Each plate included 83 participant SF single samples; one pooled OA sample; one pooled knee injury sample; five plasma calibrator samples; three plasma QC samples and three blanks per plate. The SomaScan platform quantified 7,596 synthetic DNA SOMAmers (for 7289 human targets) (Slow Off-rate Modified Aptamers [53, 54]) that bound to 6596 unique human proteins. The generated SomaScan protein quantification was securely transferred from SomaLogic to Oxford as.adat files.

Additional sample metadata, both from collecting sites (where available) and those generated in Oxford and at SomaLogic, included: sample blood staining (by visual staining defined by host investigator at time of collection); initial centrifugation of the sample; number of previous freeze thaws; the date of laboratory processing; batch and order of processing; the plate and position of the sample. These sample metadata were defined as technical confounders in our QC pipeline (S2 Table).

## Clinical data

Pseudonymised associated participant clinical data were transferred from participating sites to Oxford, linked to their consortium PIN, mapped to variables where necessary and uploaded to a REDCap database (Research Electronic Data Capture, Vanderbilt University, US) [55], hosted by the University of Oxford. Data integrity and completeness were ensured using a data dictionary, data entry constraints and a combination of automated, systematic and random checks by two of the study team.

A consortium working group oversaw all aspects of data management including definition of variables and associated data dictionary, data harmonisation and design of the database (S1 Fig). Informed by their relative clinical importance, by data availability and by iterative review, a core clinical dataset (a subset of the data dictionary) was defined: the first phenotype release "Pheno 1" (demographic data and harmonised measures of radiographic disease severity) and the second release "Pheno 2" (dichotomous and continuous harmonised patient-reported outcome measures for knee pain [56]). Characteristics of cohorts and participant samples, including details of the radiographic scoring employed (Kellgren-Lawrence [57] grading where available), are described in S1 Table with core variable definitions also provided (S2 Table and S1 File).

## Data QC approach

Methods to develop QC and data analysis pipelines prior to the primary discovery analysis were pre-defined in the Quality Assurance plan (https://www.kennedy.ox.ac.uk/oacentre/stepup-oa/stepup-oa). This QC pipeline aimed to validate methods for standardisation of the data, through a series of normalisation steps (given that SF was a non-standard matrix on SomaScan), correction for technical confounders and filtering based on pre-defined quality

thresholds for SOMAmers, proteins and samples. The approach included pre-defined data exploration, though where issues were found, these were iteratively investigated and findings used to refine the QC pipeline. This approach was informed by our prior published work [19], SomaLogic expertise, initial consortium pilot work on 435 samples previously assessed on an earlier version (4.0) of the SomaScan platform, and subsequent QC work within this dataset.

The usual SomaScan analysis pipeline for plasma involves a series of standardisation procedures to reduce nuissance variance, using plasma calibrator and plasma QC samples included on plates to reduce the effect of technical factors across samples and plates [58]. This routine standardisation of the SomaScan relative fluorescence units (RFU), adjusted by the protein's dilution factor used in the SomaScan assay (the "dilution bin"), was applied in a stepwise way, using i) hybridisation control normalisation (to remove well-to-well variation due to different rates of hybridisation between SOMAmers and fluorescence probes using spiked-in control SOMAmers); ii) plate scaling using plasma calibrators (to remove variation in overall intensity between plates); iii) median signal normalisation (to decrease variation due to total fluorescence intensity between samples); and iv) calibration (using plasma calibrator samples of known concentration to rescale each protein and reduce assay differences between runs). Each normalisation step was tested in a sequential manner.

After optimised standardisation, the R package *limma* was used to adjust proteins for a number of pre-specified continuous covariates, as described below. The batch correction method ComBat (in sva R package) was applied [59] to adjust the mean and variance of each protein for batch effects in datasets where the batch covariate, such as plate, is known [43] (again, as described below).

Based on our initial QC assessments, to quantify and adjust for differences in the contribution of intracellular proteins to the proteome, we defined an Intracellular Protein Score (IPS) for each sample *i*, as a weighted sum of log protein concentrations using the equation

$$IPS_i = \sum_p d_p C_{ip}$$

where $d_p$ is the Cohen's d for the difference in log concentration for protein *p* between paired spun and unspun samples, and $C_{ip}$ is the log concentration of protein *p* in sample *i*.

Checks of assay performance and biological validity were carried out by measuring repeatability using the pooled samples on each plate, use of metadata for prespecified technical confounders, and comparison with previously generated quantitative immunoassay data (R&D or Meso Scale Discovery, available for 60 OA and injury SF samples (without hyaluronidase treatment) for 9 overlapping proteins (S3 Table).

## Statistics and analysis

Principal Components Analysis (PCA) was used to visualise proteome-wide patterns of variation in the data, with further visualisation of Principal Components (PCs) with Uniform Manifold Approximation and Projection (UMAP) 2-dimensional plots (UMAP applied to the set of top PCs that explained >80% of total variation). Various other bioinformatic, descriptive and statistical techniques were employed to test the quality of the data. We checked:

- Inter-assay repeatability–the %CV of each protein, i.e. the ratio of the standard deviation of the concentration to the mean of the concentration within repeated samples, and the proportion of non-technical variation ($R^2$) for each protein, estimated as one minus the square of the ratio of the variance in repeated samples to the variance in non-repeated SF samples.

- The effect of freeze-thawing on normalised RFU signal (measured by %CV between repeatedly freeze-thawed and non-freeze-thawed samples).

- The effect of centrifugation on normalised RFU signal (by estimates of correlation between, and differential abundance of, proteins in unspun and spun samples, using correlation tests and paired t-tests respectively).

- Assay accuracy–comparing SomaScan normalised RFU signal with existing quantitative immunoassay data (by Pearson correlation coefficients).

- Effects of each technical confounder on standardised RFU signal (from combined cohorts). Linear regression analyses were applied to identify the most significant principal components (PCs) and proteins associated with technical confounders (Bonferonni adjusted $p<0.05$).

  In addition, SOMAmers and samples of insufficient quality were removed as follows:
  SOMAmer filtering:

- SOMAmers which were highly associated with pre-specified confounders (Bonferroni adjusted $p<0.05$) were removed (S4 Table).

- SOMAmers from non-human organisms or control SOMAmers (including Spuriomer, hybridisation control elution, deprecated, non-biotin, and non-cleavable) were excluded.

- The estimated proportion of non-technical (i.e. biological) variation $R^2$ was calculated using pooled SF samples (OA, knee injury), defined as $(1 - V_{Pooled}/V_{total})^2$, where $V_{Pooled}$ and $V_{total}$ are the variances (V) in pooled and non-pooled (including all individual) samples respectively. If $R^2$ for a given SOMAmer accounted for less than 50% of total variation in either OA or knee injury it was removed.

  Sample filtering:

- If a sample had more than 25% of protein values above or below the upper or lower limits of detection respectively, the sample was removed.

- (This was applying lower and upper limits of detection (LOD) (defined by SomaLogic), where lower LOD was: *[median concentration of blanks] + 4.9 x [median absolute deviation of blanks]*, based on three blanks (i.e. buffer only) per plate. Upper LOD was defined as 80,000 RFU).

- Identification of outliers by PCA: samples that were beyond 5 standard deviations (SDs) from the center of the principal component space (made up of PCs that explained at least 80% of variation) were removed.

- Identification of total signal intensity outliers: samples that were beyond 5 SDs from the mean in total RFU distribution were removed. Total RFU of per sample was defined as the sum of all the RFU values of that sample.

- Samples that were flagged by SomaLogic's in-house QC process were removed.

  Pearson correlation coefficients (with 95% confidence intervals) are given for correlations and all analyses were carried out in R (version 4.3.1), unless otherwise stated. For additional details, see the data analysis plan (listed in Appendix 1): https://www.kennedy.ox.ac.uk/oacentre/stepup-oa/stepup-oa.

## Results

Of 1746 unique participant samples included, tranches 1&2 were designated the Discovery analysis dataset (comprising 1045 samples), and tranches 3&4 the Replication analysis dataset (701 samples).

## Selection of each normalisation procedure

To define the optimal QC pipeline, selection of appropriate standardisation procedures was needed. These had been previously optimised by SomaLogic for plasma samples. We set out to test how well these procedures performed in all 1746 SF samples with 7596 SOMAmer features.

We measured the impact of technical variation in each normalisation step (see methods) on assay performance in a sequential manner, firstly by measuring effects on the mean %CV and the mean $R^2$ across all proteins for the pooled sample replicates across all plates (S2A and S2B Fig). Comparing each to the raw RFU data, %CVs ranged from 11.65% to 16.49% for pooled OA samples (n = 22) and 10.56% to 16.63% for pooled injury samples (n = 22). Mean $R^2$ ranged from 77.70% to 88.98% and 82.59% to 89.10% for pooled OA and injury samples respectively, suggesting a high level of repeatability. A decreasing trend in mean %CVs suggested that the routine normalisation steps improved measure repeatability with the exception of median normalisation (S2A and S2B Fig). Removing median normalisation from the standardisation procedure resulted in a mean %CV of 13.6% and a mean $R^2$ of 88.98% for pooled OA samples and %CV of 14.34% and a mean $R^2$ of 89.1% for pooled injury samples (S2A and S2B Fig).

Correlation of the SomaScan assay data with nine selected analytes measured in the same SF sample by immunoassay was generally high for most analytes (S2C and S2D Fig). However, upon normalisation steps, we saw a similar effect, with median signal normalisation reducing the correlation with these validation measurements, particularly in the injury samples (S2D Fig). Based on these combined data, we chose to use SomaLogic's existing standardisation procedures, though omitting median normalisation from our SF standardisation pipeline (i.e. employing hybridisation normalisation, plate scaling, and plate calibration using SomaLogic's plasma calibrators).

## Identification and correction of confounding factors

After conducting standardisation, principal component (PC) analysis identified one dominant component (PC1) that explained 48% of variation in the data (Fig 1A). This principal component was positively correlated with almost all proteins measured, with a median correlation coefficient of 0.70, with the highest correlations seen with low abundance proteins (Fig 1B). We were able to rule out a total protein effect, due to a low correlation (-0.039) with standard high-abundance markers such as albumin concentration. Examining the proteins that drove this signal, we found that lower protein abundance was the strongest independent predictor of correlation (S5 Table) with PC1 (p< 2.23e-308), with the next most significant predictors being whether proteins were predicted not to be secreted (p = 3.98e-10) and proteins that were identified as nuclear and not secreted (p = 1.64e-9). This led to the hypothesis that PC1 was capturing an effect of intracellular proteins, perhaps reflecting cell turnover or due to the presence of microvesicles. A strong intracellular signal was confirmed by showing that PC1 was consistently reduced in spun samples when comparing paired SF samples (from the same parent SF, n = 18) that had been split into two and either spun or left unspun immediately after joint aspiration (Fig 1C).

To quantify the contribution of intracellular proteins, we derived an Intracellular Protein Score (IPS) as the weighted sum of relative protein concentrations. For weights, we calculated a Cohen's d from the 18 paired spun and unspun samples. This score correlated very highly with PC1 (Fig 1D). We used this score in a linear regression model adjusting for the contribution of intracellular proteins, which removed the correlation between IPS and PC1 (Fig 1E),

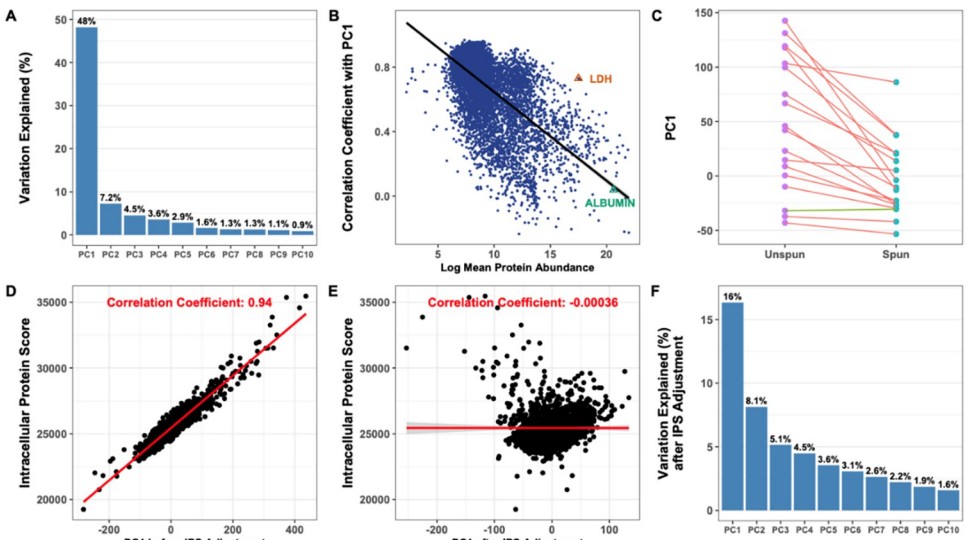

**Fig 1.** (A) Variation explained (%) by the top 10 PCs derived from the standardised log abundance proteomic data. (B) Correlation between PC1 and protein abundance, with two high-abundance proteins (albumin, a soluble serum protein, and LDH, an intracellular protein) marked. Protein abundance is calculated as the standardised RFU for each protein adjusted by the protein's dilution factor used in the SomaScan assay (the "dilution bin"). (C) Comparison of variation explained (%) by PC1 between 18 pairs of SF samples that were centrifuged (spun) or not (unspun) after aspiration and prior to freezing, with paired samples from the same participant joined by separate lines. Red lines show samples that had an increased PC1 prior to spinning, and the green line where it was decreased. Correlation between PC1 and intracellular protein score (D) before and (E) after IPS adjustment. (F) Variation explained by the top 10 PCs derived from the batch corrected and IPS adjusted log abundance proteomic data. In all cases, correlation is measured using the Pearson correlation coefficient. IPS, Intracellular Protein Score; PC, principal component; LDH, Lactate dehydrogenase.

reduced the variance explained by PC1 to 16% (Fig 1F), and removed the correlation of "non-secreted nuclear protein" with PC1 (S5 Table).

This intracellular contribution to the SF proteome did not correlate strongly with any of our pre-defined technical confounders. However, it explained a large proportion of variation in our data, which ran the risk of swamping more subtle protein signatures or molecular endotypes, if present. We thus decided to include, in addition to the standardised data set without IPS adjustment, a co-primary dataset, applying IPS adjustment to each protein as part of our STEpUP OA Data Analysis Plan (https://www.kennedy.ox.ac.uk/oacentre/stepup-oa/stepup-oa).

We also found a strong 'bimodal' signal on PC2 of the data (Fig 2A and 2B) whereby a large number of SOMAmers (N = 4030 at Benjamini–Hochberg (BH) adjusted p<0.05) were present at either very low or very high relative signal in a given sample. Further investigation showed that PC2 was highly correlated with the technical variable 'laboratory processing batch' (p<2.2E-308). Investigation of exemplar proteins displaying this behaviour showed that the bimodal signal followed sample processing order, usually (but not always) between laboratory processing batches (Fig 2C). The effect became stronger over time (Fig 2C). Re-analysing (at SomaLogic) previously laboratory processed (hyaluronidase treated) samples gave the same result. However, when new aliquots of three sequential samples which had differing bimodal status were reprocessed by the Oxford laboratory and re-analysed, all three reprocessed results had a shared bimodal status. This indicated that this was due, in some way, to our laboratory sample processing (hyaluronidase treatment) (Fig 2D). We hypothesised this might be due to sample temperature differences prior to hyaluronidase treatment, but further experiments did not corroborate this. Neither was this thought to be due to the stability of frozen hyaluronidase

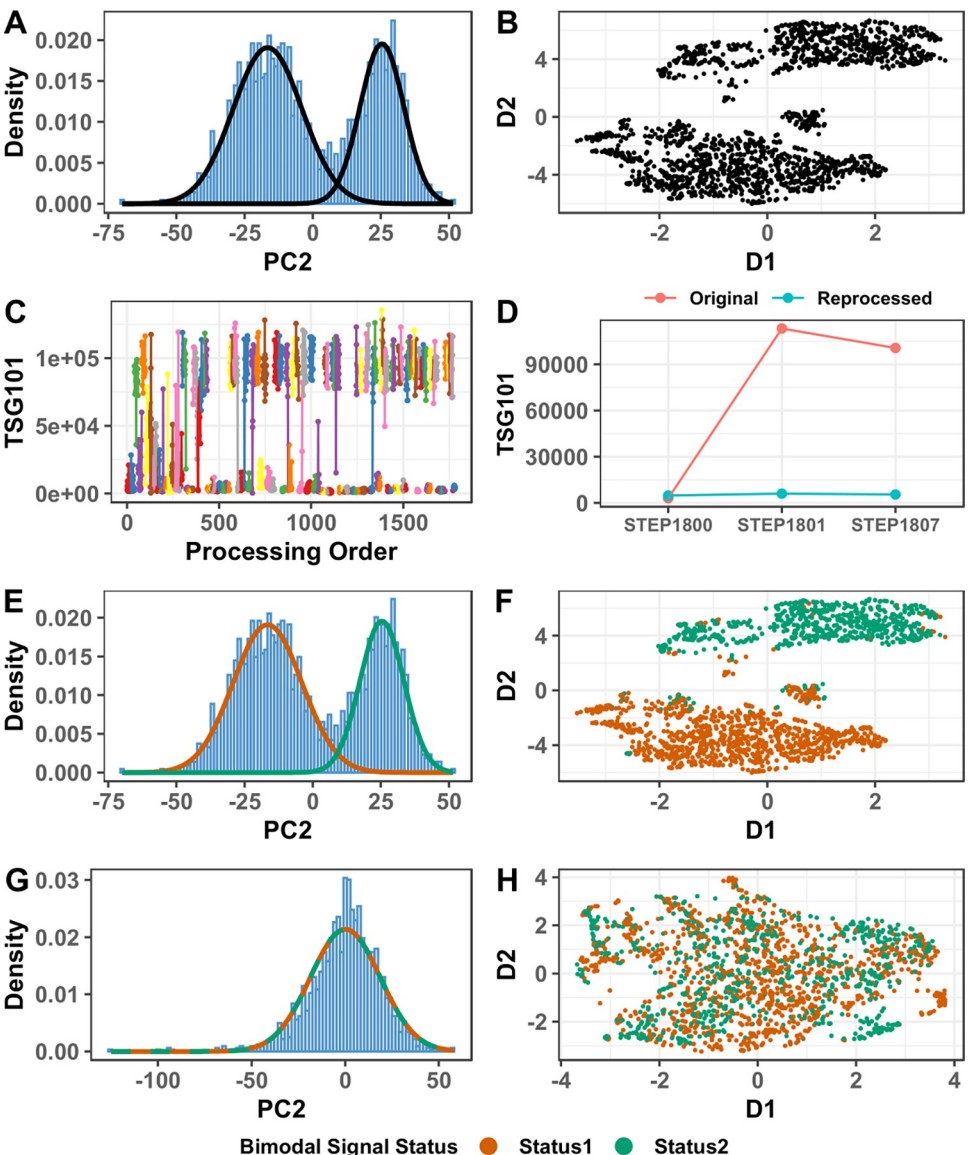

**Fig 2.** (A) Distribution of the second principal component (PC2) derived from the standardised log abundance data, showing a bimodal distribution. (B) UMAP visualisation of two reduced dimensions (D1 and D2) of the top PCs of the standardised log abundance data. (C) Example of a strongly bimodal protein measurement, TSG101, RFU (y-axis) against Oxford laboratory processing order (x-axis) and coloured by laboratory processing batch (with only points within the same processing batch connected by lines). Note that the 'flipping' between high and low signal status occurred primarily when processing batch changed, and only rarely within processing batch. This effect was particularly strong among sample batches that were processed later in processing order. (D) The same example protein measurement for three independent SF samples before (original) and after they were re-processed and re-assayed, showing that bimodal status changed after laboratory re-processing. (E) Distribution of PC2 derived from standardised log abundance data, showing the two probability density functions of the Gaussian Mixture Model used to classify samples into the two bimodal signal status groups. (F) UMAP visualisation of two reduced dimensions (D1 and D2) of the top PCs of the standardised log abundance data, colored by the inferred bimodal signal status. (G) Histogram of PC2 of the batch corrected log abundance data, with the now near-identical distributions of the two bimodal signal status groups shown as colored lines, (H) UMAP visualisation on two reduced dimensions (D1 and D2) of the top PCs of the batch corrected log abundance data, colored by the inferred bimodal signal status. RFUs, relative fluorescence units; PC, Principal Component; TSG101, Tumor susceptibility gene 101 protein; UMAP, Uniform Manifold Approximation and Projection.

enzyme as the pooled OA sample, freshly processed during each tranche with frozen stored hyaluronidase, showed little variability over time (S3 Fig).

We applied a Gaussian Mixture Model to PC2 to classify samples into high or low protein status, reflecting their bimodal signal (Fig 2E), which produced visually plausible assignments on PCA and UMAP (Fig 2E and 2F). To attempt to reduce this undesired variance, we carried out batch correction by samples' PC2 bimodal signal status using the ComBat method [59]. This correction reduced the impact of the bimodal signal considerably (Fig 2G and 2H) and was adopted into our QC pipeline.

We also discovered a significant influence of plate on a number of proteins (n = 1927) at BH adjusted p<0.05). Our samples were randomised to plate, so this was unlikely to cause significant confounding in downstream analyses, but to reduce technical variation we also applied batch correction for plate by ComBat at the same time as correcting for the bimodal signal.

We assessed the impact of these adjustments described above using the immunoassay comparison data. While the IPS-adjustment reduced the dominance of PC1, it also had a negative impact on the correlation coefficients between SomaScan and prior immunoassay results of the select analytes (S4 Fig). This was particularly evident in the injury group, where the correlation between the two measures for 4 out of 9 proteins (IL6, IL8, TGFß1, TIMP1) changed from strongly correlated to weakly or not correlated (S4B Fig). Interestingly, some of the measured cytokines (which had been selected because of their putative disease relevance) such as MCP1, IL-8 and TGFß1, correlated with the intracellular protein score (S6 Table). Batch correction for plate and bimodal signal status (as part of our optimised standardisation) was seen to have minimal impact on immunoassay agreement (S4 Fig).

## Description and reduction of pre-defined technical confounding by protein filtering

In addition to identifying badly performing SOMAmers and samples for filtering according to assay performance (see methods), we also identified filters based on pre-defined technical confounders (S2 Table).

Confounding technical factors were dealt with in different ways. Samples showed systematic biases in signal intensity plate position (S7 Table, 'Plate position'), but this was also deemed unlikely to confound downstream analyses because samples were randomised across and within plates, so its effect was not adjusted for. Blood staining (S7 Table, 'Visual blood staining') and sample volume were both drivers of IPS, but we felt that both could contain biological signals of relevance, so they were not adjusted during QC but were considered covariates in the downstream analyses. Sample age (S7 Table, 'Sample age') could introduce technical variation, therefore significantly associated proteins were removed by filtering (S4 Table). Freeze-thawing was also shown to be a potential technical confounder (S7 Table, 'Sample freeze thaw cycles'). This was investigated further.

For freeze-thawing, we had repeatedly freeze-thawed (five times per sample) one aliquot each of the pooled OA and the pooled injury samples. This had an effect, particularly in the injury samples. However, the majority (77%) of proteins retained a good %CV (<20%) even after five freeze-thaws (S3 Fig), suggesting that such samples remained usable. Technical variation brought about by freeze-thaw was nonetheless adjusted for by filtering out significantly associated proteins following Bonferroni correction (S4 Table).

## Assessment of centrifugation effect on protein measurements

Although most of the samples had been centrifuged prior to initial storage as per our eligibility criteria, 240 samples included in the replication analysis were unspun. In anticipation of this

we assessed further the impact of centrifugation on the 18 pairs of samples that had either been spun or left unspun at time of collection. We compared the SomaScan data to identify proteins that changed upon centrifugation (S5 Fig). The effect of centrifugation on the data depended on whether the data were adjusted for IPS. The unadjusted data showed that centrifugation status was a major driver of variation across the paired samples, with a significant correlation with PC1 (S5A Fig, paired t-test p = 0.0066), but after adjusting for IPS, the top PCs were no longer driven by spun status (S5B Fig, paired t-test p = 0.2089). Centrifugation impacted the concentration of a large number of individual proteins in both IPS-unadjusted (n = 5638, 74%, at BH adjusted p<0.05) and, to a lesser extent, IPS-adjusted data (n = 3731, 49%, at BH adjusted p<0.05), although the majority of proteins were significantly correlated between the paired spun and unspun samples (n = 6402, 85% in unadjusted data and n = 4558, 60% in IPS adjusted data, S5C and S5D Fig respectively).

We concluded that spun and unspun samples were comparable (in that they captured similar information), but that any analysis that included both types together would need to adjust for systematic shifts in abundance and the small numbers of uncorrelated proteins. In our discovery and replication analysis plans relating to our primary analysis, only spun samples are therefore considered, with unspun samples used for secondary sensitivity analyses.

## Effect of blood staining on protein measurements

A subset of samples had information on blood staining, graded by visual inspection at the time of joint aspiration, prior to centrifugation. As shown in S7 Table, the presence of blood measured in this way was a significant driver of protein variation. It was also a potential biological driver as haemarthrosis is common after significant joint injury and is known to be pro-inflammatory and associated with persisting knee symptoms [60–62]. Visual blood staining could reflect presence of either intact or lysed red blood cells. The analyte haemoglobin A (HBA) correlated reasonably well with visual blood staining grade prior to adjustment for IPS (S6A Fig), and less so after adjustment for IPS (S6B Fig). The log concentration of HBA relative abundance level (without IPS adjustment) was subsequently used as a measure of blood content, as a covariate in downstream analyses.

## Validation of data quality after QC

Following application of filters, 1720 samples and 6290 SOMAmers (features) remained. The total numbers of samples and proteins filtered out are shown in S4 Table. After filtering, median %CV of pooled OA and injury samples remained relatively unchanged at 11.25 and 12.42 respectively (S7 Fig). An overview of the end-to-end data processing and quality control pipeline, from raw data to final filtered data, is shown in Fig 3.

The association of all these variables with the top 10 PCs of the standardised, bimodal signal corrected data after filtering is shown in S7 Table. The strongest associations in the IPS adjusted, filtered data are shown in Fig 4. IPS adjusted non-filtered, and non-IPS adjusted data are shown in S8 Fig.

Finally, we visualised the different diagnostic subgroups (OA, joint injury, inflammatory control, disease-free control) on UMAPs of the standardised, corrected and filtered data, with and without IPS adjustment (Fig 5A and 5B respectively). Both datasets showed clustering of knee injury and OA cases in overlapping but distinguishable areas of high-dimensional proteomic space, though the smaller groups (disease-free controls and inflammatory controls) were more evenly distributed. Inflammatory controls tended to segregate with acute knee injury samples. These patterns were also reflected at the PC level (Fig 4F and S9 Fig).

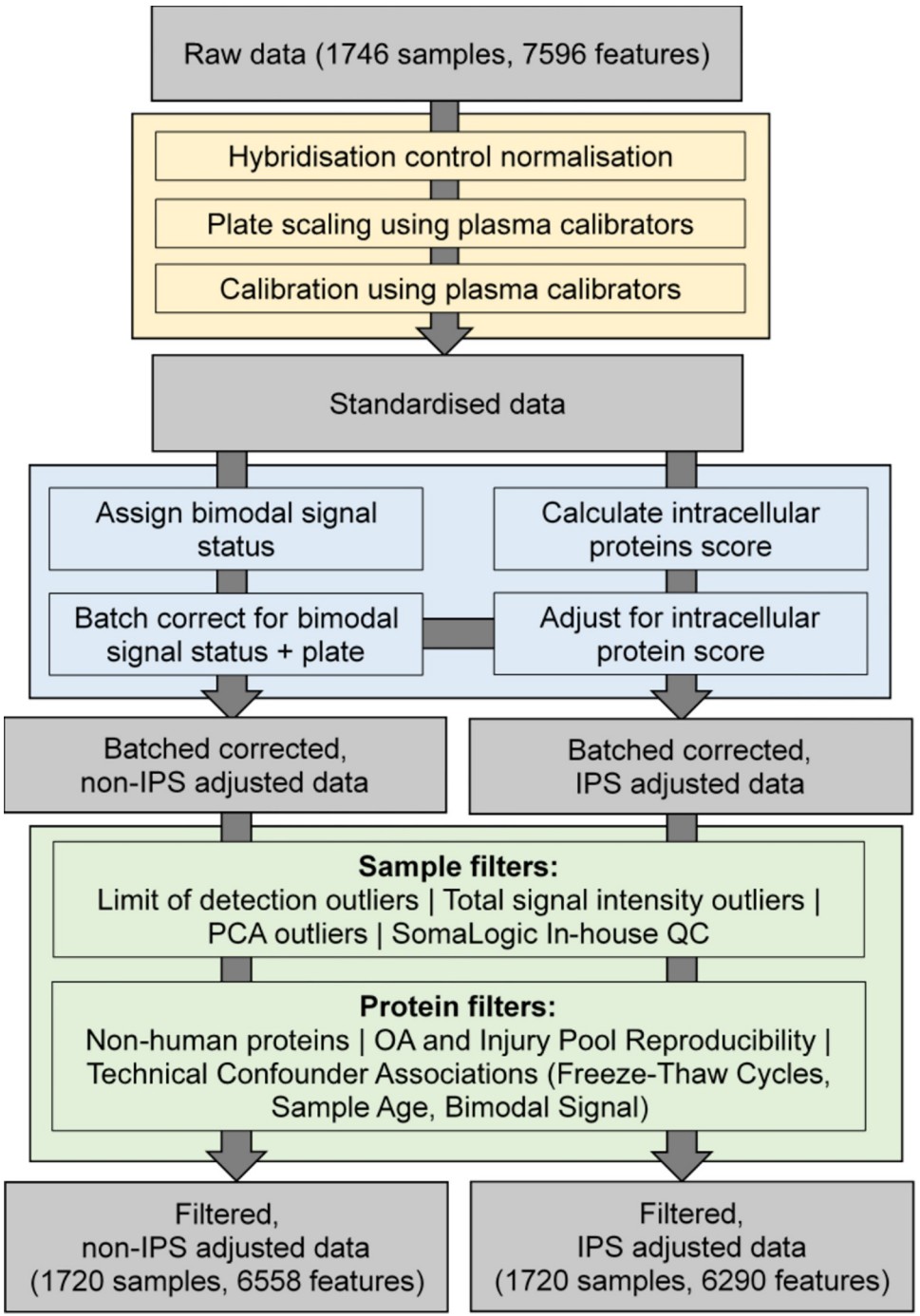

**Fig 3. Overview of the final data processing and quality control pipeline for synovial fluid SOMAscan data used by the STEpUP OA consortium, broken down into three stages: Standardisation (yellow box), technical confounder correction (blue box) and filtering (green box).** More details on filtering thresholds, and the number removed by each filter, can be found in S4 Table.

## Discussion

In this initial report from the STEpUP OA consortium, we describe a comprehensive evaluation of the overall performance of the SomaScan assay for knee SF for the first time. We

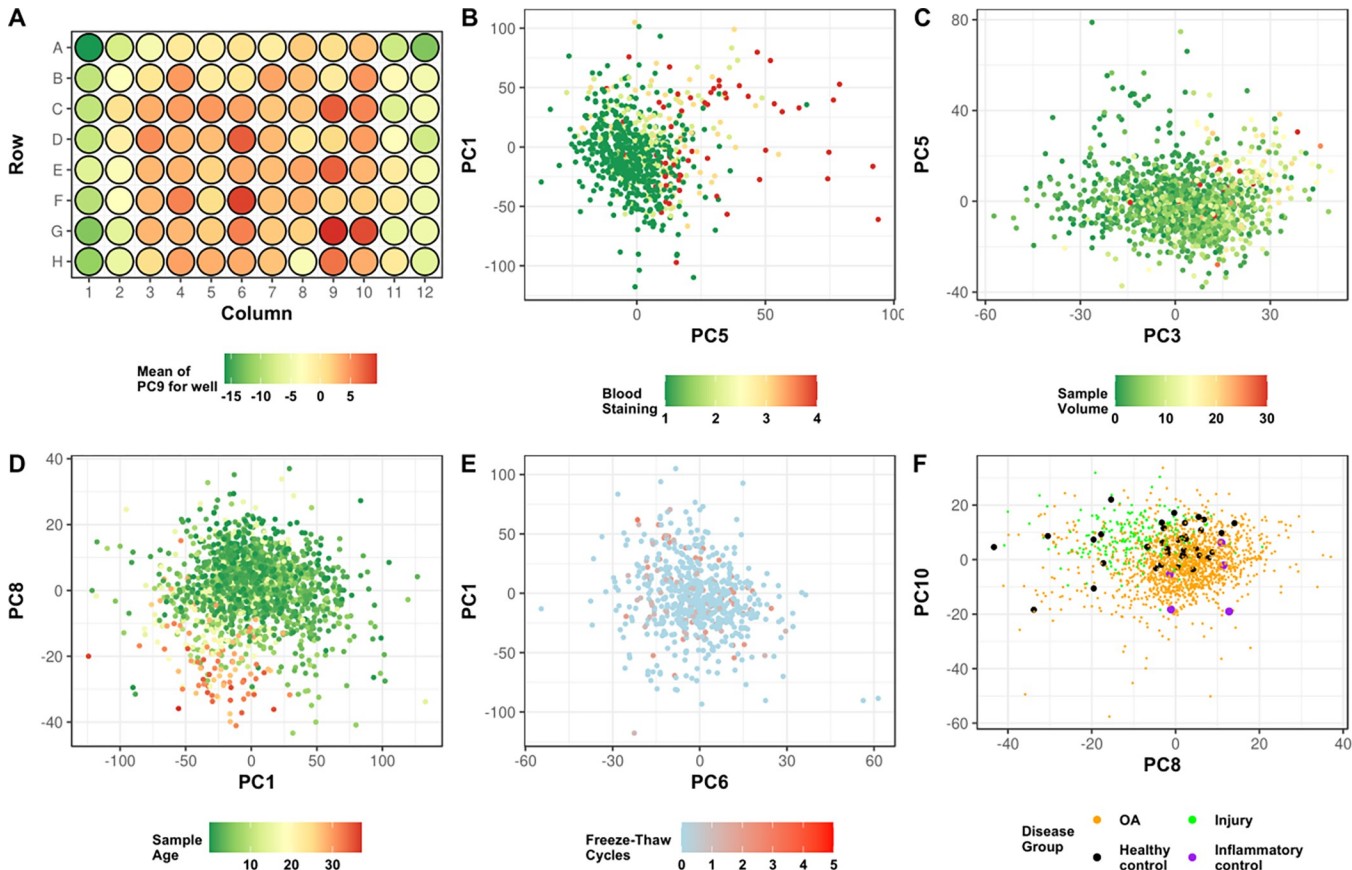

**Fig 4. Visualisation of selected predefined confounders against select principal components of the batch corrected, filtered, IPS adjusted data.** (A) The average value of PC9 (most strongly associated with plate position) by sample well position, (B-F) visualisation of the two PCs most strongly associated with each confounder, coloured by confounder value. Pre-defined confounders shown are (B) blood staining grade of sample after aspiration assessed by visual inspection, (C) volume of sample taken during aspiration, (D) age of the sample in years, measured from aspiration to sample processing at Oxford, (E) the number of times the sample was thawed and re-frozen before sample processing at Oxford, (F) the disease group of the sample (osteoarthritis [OA], acute knee injury [Injury], healthy control, inflammatory arthritis control).

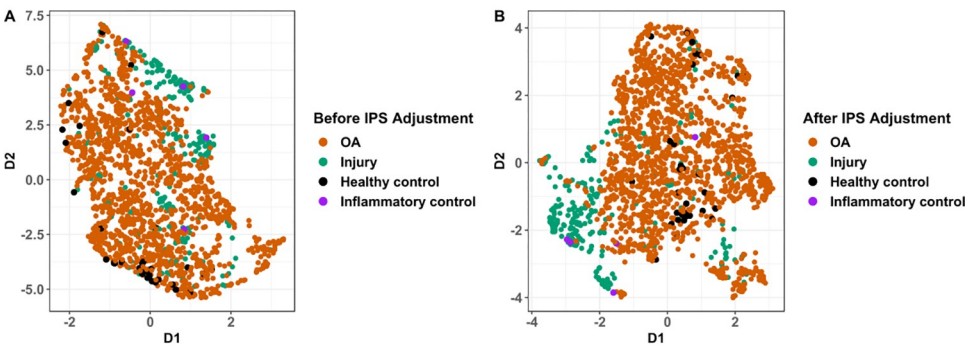

**Fig 5.** UMAP visualisation of two reduced dimensions (D1 and D2) of the top PCs of the log abundance data with (A) and without (B) IPS adjustment followed by filtering, coloured by disease group. These groups were osteoarthritis (OA, acute knee injury (injury), healthy controls, inflammatory arthritis controls. UMAP, Uniform Manifold Approximation and Projection.

address a series of data processing and analysis challenges that arise from proteomic quantification of SF using this technology. Based on our investigations, we propose an optimal standardisation procedure for SF and an assessment of the quality of the protein data using pre-defined approaches. Our aim was to justify the best approach to minimise technical variation while maintaining biological variation in the data, and thus to develop a pipeline that could be applied to downstream analyses within STEpUP OA and in subsequent proteomic analyses of SF by others.

We identified a number of technical confounders, which all affected, to a great or lesser extent, the data structure. These included factors that related to the SF sample e.g. its age, number of freeze-thaws, as well as potential confounders that could arise during the sample processing e.g. plate and date of processing. Because we had randomised the samples to and within plate, we were also able to identify plate position and laboratory processing batch as additional confounding factors. Each of these was controlled for by adjustment or filtering (either sample or SOMAmer). Whilst our intention was to perform the primary analysis of STEpUP OA only in spun SF sample data, we also included a number of unspun samples to test the generalizability of these to the larger dataset. In doing so we calculated that more stringent filtering would be needed when studying unspun, or mixed spun/unspun SF collections. We chose not to correct for blood staining (or HBA) or sample volume as we felt that these could reflect important biological variation.

The finding that a large proportion of variance in the data was driven by intracellular protein was unexpected. This could have arisen as a result of technical confounding following contamination of the SF samples by cells e.g. by a failure to remove cells fully by centrifugation, or by cell lysis of those cells at the time of aspiration e.g. by delay in spinning sample down. It could also be a true reflection of cellular turnover within the joint as part of the disease process e.g. of infiltrating immune cells or native connective tissues. It could also reflect protein carried within microvesicles that are known to be increased in joint disease and which drive biology within and between joint tissues [63, 64]. These possibilities are currently being explored. There is a concern that if the IPS reflects true biology, then adjusting for it may compress true signals within the data. This is consistent with the reduction in correlation with immunoassay seen in the injury samples. On the other hand, subtle structures within the data that reflect true molecular endotypes might be masked without removal of this signal. For this reason, the relevance of the IPS to clinical parameters and endotype clusters will be addressed alongside one another in the primary analysis of STEpUP OA.

Our data demonstrate that, when properly processed, SomaScan produces repeatable and accurate quantification of SF proteins which can (as a quality check, not a diagnostic one) broadly separate different clinical groupings. Its quantitation quality is at least comparable/superior to other 'non-standard' matrices e.g. urine/cerebrospinal fluid previously studied on this platform [65–67]. Our repeatability measures in pooled synovial fluid samples, with median %CVs of 11.25% for OA sample replicates and 12.42% for injury sample replicates, were higher than have been reported for plasma and serum samples measured using SomaScan technology, where %CVs of 5% or less are observed [24]. However, our correlation with immunoassays (median coefficient 0.81 for OA and 0.92 injury respectively) are as good or better than are typically observed in blood samples [68, 69]. Our data quality metrics are comparable with those based on mass spectrometry or other quantitative immunoassays, where %CVs of 10% or greater are recorded [20, 70]. Compared with these technologies our data has higher dynamic range and sensitivity (as the technology can assay proteins that are at very high as well as low abundance), noting only proteins on a pre-defined (though very large) protein list are included. Free nucleic acids in biological fluids such as SF could potentially interfere with the SomaScan assay. However, this is unlikely as the aptamers used for protein

identification, which are chemically modified DNA molecules, are high affinity for that protein allowing low affinity binders to be competed off through application of an unlabeled polyanion competitor [71]. It might have been possible to reduce interference further by pre-treating SF with DNAse or RNAse, but this is not considered standard for SomaScan analysis [72, 73].

There are several limitations of this large study. The consortium collection was highly heterogeneous, gathered from seventeen different studies, varying in disease severity and phenotype, across several decades and from a number of countries without a unified pre-specified sample processing protocol. This made distinguishing technical and biological variation difficult (as was the case for the intracellular protein score, which could reflect either variation in joint biology or variation in sampling handling). Intrinsic protease activity within the fluids may have contributed to confounding associated with sample age. Addition of protease inhibitors at the time of collection could be something to consider in future sample collections, but is generally not the norm. A further limitation is that we analysed only a single matrix (SF) although this likely reflects activity in multiple joint tissues. The lack of paired cartilage/synovium/bone in STEpUP OA prevents us performing a direct integrated analysis using RNAseq, for example, although it may be possible to extrapolate this from other existing datasets. Other proteome-wide technologies (such as LC-MS/MS, OLINK [74]) could provide further validation on protein patterns within OA SF. Paired plasma is also available for many individuals in STEpUP OA but is yet to be analysed. From previous experience we would predict that this matrix would show low concordance with SF [18, 75]. Others have used SomaScan to explore the plasma proteome in OA, identifying diagnostic and prognostic biomarkers, though this study did not study paired SF samples [50].

In summary, we present an evidence-based methodology pipeline for large scale proteomic analysis on the SomaScan platform of SF, which has the potential to be a critical matrix for discovery science and clinical translation in OA. Our next step, the primary analysis of this dataset, seeks to answer definitively whether there are distinct discernible molecular endotypes, by unsupervised k means clustering, in this common, yet poorly understood disease.

## Supporting information

**S1 Fig. Consortium structure, as working groups.** Distinct working groups oversaw key activities according to pre-defined Terms of Reference (available on request). TV, Tonia Vincent; FW, Fiona Watt; AV, Ana Valdes; LJD, Luke Jostins-Dean; RM, Rose Maciewicz. (TIF)

**S2 Fig.** Assessment of the effects of each standardisation step on (A) mean %CV and (B) mean $R^2$ across all proteins for pooled sample replicates, stratified by OA and acute knee injury respectively. Assessment of Pearson correlation coefficients between protein expression in samples measured by the SOMAscan platform and by prior immunoassay for nine select proteins across normalisation steps for (C) OA and (D) acute knee injury. The normalisation steps included hybridisation normalisation (HN), plate scaling using plasma calibrators (PS), median signal normalisation (MN) and calibration using plasma calibrators (PC). Correlation between the RFUs (SOMAscan) and absolute concentrations for the nine proteins across the two methods are shown. RFUs, relative fluorescence units; %CV, % coefficient of variation; osteoarthritis, OA; Activin A, Inhibin beta A chain; FGF2, Fibroblast growth factor 2; IL6, Interleukin-6; IL8, Interleukin-8; MCP1, C-C motif chemokine 2; MMP3, Stromelysin-1; TGFβ1, Transforming growth factor beta-1; TIMP1, Tissue inhibitor of metalloproteinase 1; TSG6, Tumor necrosis factor-inducible gene 6. (TIF)

**S3 Fig.** Assessment of assay repeatability using pooled samples of synovial fluid from participants with (A) knee OA and (B) acute knee injury, measured by the coefficient of variation (% CV). These include the repeatability of the standard processed pooled samples included on every plate ('Sample Repeats'), pooled samples which had been repeatedly freeze-thawed ('Freeze Thaw') prior to processing and an OA pool aliquot that had been freshly enzyme digested with stored hyaluronidase during each of the 2nd, 3rd and 4th tranches of sample processing (done for the OA pool only) ('Reprocessed'). Dotted vertical lines show the maximum %CV for 80% of proteins for each group.
(TIF)

**S4 Fig.** Correlation between SOMAscan relative frequency abundance (RFU) and abundance measured using orthogonal immunoassays for 9 selected proteins at different stages of SOMAscan data processing, for (A) osteoarthritis and (B) acute knee injury samples. Correlation was measured using the Pearson correlation coefficient. Raw data refers to the raw RFUs without any processing, optimised standardisation was the data standardised using our selected optimal normalization steps (S2 Fig), processed without IPS adjustment refers to data that has been batch corrected for bimodal signal status and plate but not IPS adjusted, and processed with IPS adjustment refers to samples that have undergone both batch correction and IPS adjustment. IPS, Intracellular Protein Score; Protein name abbreviations as in S2 Fig.
(TIF)

**S5 Fig.** Top 2 principal components of (A) non-IPS adjusted and (B) IPS adjusted log RFU of the 18 pairs of centrifuged (spun) and non-centrifuged (unspun) SF samples. Samples are coloured by spin status and paired samples are linked by lines. Measures of differential abundance (Cohen's d) and Pearson correlation coefficient (rho) between spun and unspun samples for (C) non-IPS adjusted and (D) IPS adjusted log RFU. Samples are coloured depending on their significance (Benjamini-Hochberg adjusted $p < 0.05$) on the two measures: Different Means corresponds to a significant difference in means in a paired t-test and Correlated corresponds to a significant correlation in a Pearson correlation test. IPS, intracellular protein score; PC, principal component; SF, synovial fluid; RFU, relative fluorescence unit.
(TIF)

**S6 Fig.** Boxplots showing the correlation between visual blood staining grade of SF at the time of sample collection and the blood analyte, HBA, in non-IPS adjusted data in (A) all samples, (C) OA samples and (E) acute knee injury samples, and in IPS adjusted data in (B) all samples, (D) OA samples and (F) acute knee injury samples. Spearman correlation coefficients measuring rank-based correlation considering visual blood staining as an ordinal variable are shown. 443 OA samples had blood staining grade 1 (no blood detected, 75% among the 588 total samples with blood staining records). HBA, haemoglobin A; IPS, intracellular protein score; SF, synovial fluid.
(TIF)

**S7 Fig.** (A) Assessment of assay repeatability after optimised quality control procedures measured using the cumulative distribution of the coefficient of variation (%CV) on pooled OA samples (OA Sample Repeats) and pooled acute knee Injury samples (Injury Sample Repeats) separately. 80% of proteins had a %CV less than 16.85% and 17.57% in the OA and acute knee injury pools (blue and red dotted lines respectively). (B) The proportion of variation that was estimated to be non-technical, measured by $R^2$ for OA and acute knee injury sample repeats separately. 80% of proteins had $R^2$ values greater than 88.27% and 84.33% in the OA and knee injury pools (blue and red dotted lines respectively).
(TIF)

**S8 Fig.** Visualisation of pre-defined technical confounders by select principal components of the (A) non-filtered IPS adjusted (B) filtered non-IPS adjusted and (C) non-filtered non-IPS adjusted data. Visualisation of the two PCs most strongly associated with each confounder (colours correspond to confounder value). Confounders include plate position (mean of PC8), blood staining grade of sample (which was performed immediately after aspiration from the joint by visual inspection), volume of sample taken during aspiration, age of the sample in years from aspiration to processing, the number of times the sample had been thawed and re-frozen, the disease group of the sample (knee OA, acute knee injury, healthy control, inflammatory arthritis control). The association between each PC and confounder is shown in the S7 Table.
(TIF)

**S9 Fig.** Pairwise scatter plots (off-diagonal) and histograms (diagonal) of the top five principal components of standardised log abundance, (A) before and (B) after batch correction for plate and bimodal signal status, coloured by bimodal signal status. Batch correction effectively removed the effect of bimodal signal status on the top PCs.
(TIF)

**S1 Table. Characteristics of cohorts and participant samples.**
(DOCX)

**S2 Table. Core clinical phenotype data used for quality control and downstream analyses.**
(DOCX)

**S3 Table. Summary of proteins measured by immunoassay used to assess accuracy of SomaScan data.**
(DOCX)

**S4 Table. Summary of sample and protein filters.**
(DOCX)

**S5 Table. Predictors of the strength of correlation between protein abundance and PC1.**
(DOCX)

**S6 Table. Correlation between Intracellular protein score and protein abundance measured by immunoassay.**
(DOCX)

**S7 Table. Associations between technical confounders and top 10 PCs.**
(DOCX)

**S1 File. Supplementary methods.**
(DOCX)

## Acknowledgments

We would like to express our gratitude and thanks to all cohorts and their participants who contributed samples to STEpUP OA. We are grateful for the support from Floris Lafeber and Simon Mastbergen (Utrecht Medical Centre). This work was also supported by the NIHR Oxford Biomedical Research Centre (BRC) and the NIHR Nottingham BRC. The views expressed are those of the authors and not necessarily those of the NHS, the NIHR or the Department of Health. Tissue samples and/or data obtained from the Oxford Musculoskeletal Biobank were collected with informed donor consent in full compliance with national and

institutional ethical requirements, the UK Human Tissue Act, and the Declaration of Helsinki (HTA Licence 12217 and Oxford REC C 09/H0606/11). We thank the Oxford Knee Surgery Team including Andrew Price, William Jackson and Nicholas Bottomley and our centre tissue coordinators Louise Hill and Katherine Groves who coordinated this study. We thank Charlotte Kerr for her administrative support of the consortium at large.

The STEpUP OA Consortium author block includes: University of Nottingham: Ana M. Valdes, David A. Walsh, Michael Doherty, Vasileios Georgopoulos; Lund University: Staffan Larsson, L. Stefan Lohmander, André Struglics; University of Cambridge: Brian D.M. Tom, Laura Bondi; University of Toronto: Mohit Kapoor, Rajiv Gandhi, Anthony Perruccio, Y. Raja Rampersaud, Kim Perry; University of Manchester: Tim Hardingham, David Felson; University of Oxford: Tonia L. Vincent, Thomas A. Perry, Luke Jostins-Dean, Yun Deng, Vicky Batchelor, Jennifer Mackay-Alderson, Gretchen Brewer, Rose M. Maciewicz, Brian Marsden, Nigel K. Arden, Philippa Hulley, Andrew Price, Stefan Kluzek, Megan Goff, Vinod Kumar, James Tey; Imperial College London: Fiona E. Watt, Andrew Williams, Artemis Papadaki; University College Maastricht: Tim J. Welting, Pieter Emans, Tim Boymans, Liesbeth Jutten, Marjolein Caron, Guus van den Akker; University of Western Ontario: C. Thomas Appleton, Trevor B. Birmingham, J. Daniel Klapak; Biosplice: Sarah Kennedy, Jeymi Tambiah; Fidia: Devis Galesso, Nicola NK; SomaLogic: Joe Gogain, Darryl Perry, Anna Mitchel, Ela Zepko; Novartis: Sophie Brachat, Joanna Mitchelmore, Juerg Gasser, Lori Jennings; UCB: Waqar Ali.

TLV directs the Centre for OA pathogenesis (grant numbers 21612 and 20205) and has additional grant support from Versus Arthritis, the European Research Council, the Medical Research Council and FOREUM. LJD is supported by a Wellcome trust fellowship grant 208750/Z/17/Z and Kennedy Trust for Rheumatology Research for the present manuscript. LJD is also supported by grants from the MRC and the Helmsley Charitable Trust. FEW is supported by a UKRI Future Leaders Fellowship (MRC number: MR/S016538/1 and MR/S016538/2). FW, NKA and SK are members of the Centre for Sport, Exercise and Osteoarthritis Research Versus Arthritis (grant number 21595). MK is supported by grants from CIHR, NSERC, The Arthritis Society Canada, Krembil Foundation, CFI, Canada Research Chairs program, and has received support from the University Health Network Foundation, Toronto for the present manuscript. TJW is supported by grants from NWO-TTW Perspectief (#P15-23), Stichting de Weijerhorst and ReumaNederland (LLP14) for the present manuscript, and is a shareholder of Chondropeptix BV. CTA is supported by the Canadian Institutes of Health Research, Western University Bone and Joint Institute, and the Academic Medical Organization of Southwestern Ontario for the present manuscript. BDMT is supported through the United Kingdom Medical Research Council programme (grant MC UU 00002/2). For the purpose of open access, the authors have applied a Creative Commons Attribution (CC BY) license to any Author Accepted Manuscript version arising. LB is supported by grants from Kennedy Trust for Rheumatology Research (grant number 171806) and UK Medical Research Council (grant MC UU 00002/2). DAW is supported by grants from Pfizer Ltd, UCB Pharma, Orion Corporation, GlaxoSmithKline Research and Development, and Eli Lilly and Company, Versus Arthritis, UKRI, Nuffield Foundation.

## Patient and public involvement statement

People with lived experience of osteoarthritis have been involved in the design of this project. A patient research panel was involved in discussing and inputting on the STEpUP OA project in February 2020 (invited to the Centre for Osteoarthritis Pathogenesis Versus Arthritis in Oxford, as part of its involvement activities). Aspects relevant to the development of the project were further discussed with the panel in July 2022. The working groups for the consortium

include one focused on patient involvement and engagement. A lay summary is included in the appendix of our publicly available analysis plan. A short video about the project was produced and is available on our website: https://www.kennedy.ox.ac.uk/oacentre/stepup-oa/stepup-oa. In addition, the various constituent cohorts contributing to STEpUP OA also typically have lay or patient members on their steering committees.

## Author Contributions

**Conceptualization:** Tonia L. Vincent, Fiona E. Watt, Luke Jostins-Dean.

**Data curation:** Yun Deng, Thomas A. Perry, Brian Marsden, Vinod Kumar, Fiona E. Watt, Luke Jostins-Dean.

**Formal analysis:** Yun Deng, Thomas A. Perry, Rose A. Maciewicz, Joanna Mitchelmore, Darryl Perry, Brian D. M. Tom, Laura Bondi, Luke Jostins-Dean.

**Funding acquisition:** Tonia L. Vincent.

**Investigation:** Tonia L. Vincent, Fiona E. Watt, Luke Jostins-Dean.

**Methodology:** Yun Deng, Thomas A. Perry, Philippa Hulley, Rose A. Maciewicz, Joanna Mitchelmore, Darryl Perry, Sophie Brachat, Brian D. M. Tom, Laura Bondi, Vicky Batchelor, Jennifer Mackay-Alderson, Tonia L. Vincent, Fiona E. Watt, Luke Jostins-Dean.

**Project administration:** Thomas A. Perry, Brian Marsden, Vicky Batchelor, Jennifer Mackay-Alderson, Vinod Kumar, Tonia L. Vincent, Fiona E. Watt, Luke Jostins-Dean.

**Resources:** C. Thomas Appleton, Stefan Kluzek, Nigel K. Arden, Mohit Kapoor, L. Stefan Lohmander, Tim J. Welting, David A. Walsh, Ana M. Valdes, Fiona E. Watt.

**Software:** Brian Marsden, Vinod Kumar.

**Supervision:** Tonia L. Vincent, Fiona E. Watt, Luke Jostins-Dean.

**Writing – original draft:** Yun Deng, Thomas A. Perry, Tonia L. Vincent, Fiona E. Watt, Luke Jostins-Dean.

**Writing – review & editing:** Yun Deng, Thomas A. Perry, Philippa Hulley, Rose A. Maciewicz, Joanna Mitchelmore, Darryl Perry, Staffan Larsson, Sophie Brachat, André Struglics, C. Thomas Appleton, Stefan Kluzek, Nigel K. Arden, David Felson, Brian Marsden, Brian D. M. Tom, Laura Bondi, Mohit Kapoor, Vicky Batchelor, Jennifer Mackay-Alderson, Vinod Kumar, L. Stefan Lohmander, Tim J. Welting, David A. Walsh, Ana M. Valdes, Tonia L. Vincent, Fiona E. Watt, Luke Jostins-Dean.

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
