## [Decision Letter · Decision Letter 0]

7 Dec 2023

PONE-D-23-35180Development of molecular endotype discovery from synovial fluid of individuals with knee osteoarthritis: the STEpUP OA ConsortiumPLOS ONE

Dear Dr. Perry,

Thank you for submitting your manuscript to PLOS ONE. After careful consideration, we feel that it has merit but does not fully meet PLOS ONE’s publication criteria as it currently stands. Therefore, we invite you to submit a revised version of the manuscript that comprehensively addresses the points raised by the reviewers. Since we only received comments from one reviewer, we will send the paper out for full review once you have submitted the amended paper. During resubmission, we recommend that you provide additional suggested reviewers who are willing to provide unbiased but constructive comments to expedite the second round of editorial review.

We look forward to receiving your revised manuscript.

Kind regards,

Andre van Wijnen

Academic Editor

PLOS ONE

Journal Requirements:

"The study was supported by Kennedy Trust for Rheumatology Research (grant number: 171806), Versus Arthritis (grant number: 22473), Centre for OA Pathogenesis Versus Arthritis (grant numbers: 21621, 20205), Galapagos, Biosplice, Novartis, Fidia, UCB, Pfizer (non consortium member) and Somalogic (in kind contributions). "

"YD, TAP, PH, SL, AS, NKA, DF, BM, AMV, SK, VB, JMA and VK declare no conflicts of interest. FW has received consultancy fees from Pfizer, and has a leadership role at the Medical Research Council (panel member) and Osteoarthritis and Cartilage (Associate Editor). LSL has received consultancy fees from Arthro Therapeutics AB, and is an advisory board member of AstraZeneca (non consortium member).  LJD has received consultancy fees from Nightingale Health PLC. TLV has no conflicts to declare with the exception of grant income for STEpUP OA from industry partners (see above). RAM is a shareholder of AstraZeneca. SB and JM are employees and shareholders of Novartis (consortium members). MK has received support for attending the Gordon Research Conference, OARSI meeting, International Cartilage Repair Society, Munster University, is a board member of the Dutch Arthritis Society (Chair of Visitation Board), and has a leadership role at Osteoarthritis Research Society International (Board of Directors Member). CTA has received consultancy fees from Novartis, and has received honoraria for educational purposes also from Novartis. DAW has received consultancy fees from GlaxoSmithKline plc, AKL Research & Development Limited, Pfizer Ltd, Eli Lilly and Company, Contura International, and AbbVie Inc, has received honoraria for educational purposes from Pfizer Ltd and AbbVie Inc, is a board member of UKRI (Director) and Versus Arthritis Advanced Pain Discovery Platform. "

8. We notice that your supplementary [Figure S1-S9] are included in the manuscript file. Please remove them and upload them with the file type 'Supporting Information'. Please ensure that each Supporting Information file has a legend listed in the manuscript after the references list.

*Comments from PLOS Editorial Office: *

*1) We note that one of the reviewers expressed concerns about the quality of the images. Although the figures appear to be of poor quality when viewed as part of the pdf, the individual figure files are of a good quality. The original figures can be accessed by clicking on the link at the top of each page with a figure to download the file. Therefore, you do not need to produce new versions of the figures at this time, although please see note 8 above.*

*2) Although there are comments from 2 reviewers below, reviewer 2 felt unable to fully assess your submission. Please ignore the comments provided by reviewer 2.*

Additional Editor Comments:

Reviewer#1:

Data availability- the data from the main study (proteomics) is not open access data. The mention of a fee to access needs more detail.

Abstract

The use of abbreviations should be removed in the abstract for easier reading. Indeed, some abbreviations (synovial fluid) are not even defined.

Introduction

Minor

Check abbreviations are defined the first time used (eg FDA).

Mixture of UK and US English, be consistent and use UK English.

Methods

More details on the SF preparation from collection to storage are required. What was the time from collection to centrifugation, were samples subaliquot, where samples collected into low protein bind tubes? How long were samples stored at -80 for prior to analysis? Was the supernatant removed following centrifugation? Were samples kept in a fridge at all times? Was a cold centrifuged used? Were samples with blood contamination excluded? Were samples were collected following joint lavage what buffer was used and what were the protocols?

Details of why certain criteria were used for filtering should be given from line 238-251.

Results

I cannot assess the remaining manuscript as the quality of the images is too of too poor resolution.

Reviewer#2: No comments provided.

Reviewers' comments:

Reviewer's Responses to Questions

**Comments to the Author**

1. Is the manuscript technically sound, and do the data support the conclusions?

Reviewer #1: Yes

Reviewer #2: Yes

2. Has the statistical analysis been performed appropriately and rigorously? 

Reviewer #1: I Don't Know

Reviewer #2: I Don't Know

3. Have the authors made all data underlying the findings in their manuscript fully available?

Reviewer #1: No

Reviewer #2: Yes

4. Is the manuscript presented in an intelligible fashion and written in standard English?

Reviewer #1: No

Reviewer #2: Yes

5. Review Comments to the Author

Reviewer #1: Data availability- the data from the main study (proteomics) is not open access data. The mention of a fee to access needs more detail.

Abstract

The use of abbreviations should be removed in the abstract for easier reading. Indeed, some abbreviations (synovial fluid) are not even defined.

Introduction

Minor

Check abbreviations are defined the first time used (eg FDA).

Mixture of UK and US English, be consistent and use UK English.

Methods

More details on the SF preparation from collection to storage are required. What was the time from collection to centrifugation, were samples subaliquot, where samples collected into low protein bind tubes? How long were samples stored at -80 for prior to analysis? Was the supernatant removed following centrifugation? Were samples kept in a fridge at all times? Was a cold centrifuged used? Were samples with blood contamination excluded? Were samples were collected following joint lavage what buffer was used and what were the protocols?

Details of why certain criteria were used for filtering should be given from line 238-251.

Results

I cannot assess the remaining manuscript as the quality of the images is too of too poor resolution.

Reviewer #2: Nnnnnnnnnnnnnnnnnnnnnnnnnnnnnnnnnnnnnnnnnnnnnnnnnnnnnnnnnnnnnnnnnnnnnnnnnnnnnnnnnnnnnnnnnnnnnnnnnnnnnnnnnnnnnnnnnnnnnnnnnnnnnnnnnnnnnnnnnnnnnnnnnnnnnnnnnnnnnnnnnnnnnnnnnnnnnnnnnnnnnnnnnnnnnnnnnnnnnnnnnn

6. PLOS authors have the option to publish the peer review history of their article (what does this mean?). If published, this will include your full peer review and any attached files.

Reviewer #1: No

Reviewer #2: No

---

## [Author Response · Author response to Decision Letter 0]

9 Feb 2024

PLOS ONE Decision: Revision required [PONE-D-23-35180] - [EMID:a292826ab746f870]

Response to reviewer comments

We thank the reviewers and editorial team for their helpful comments, questions and suggestions and detail below a point-by-point response to these. Page numbers have been given with reference to the resubmitted tracked changes manuscript and supplementary materials. 

Response to Editorial points:

Author Response: Thank you for highlighting these formatting requirements.

Author Action: These have now been corrected throughout the text and we hope the manuscript structure now aligns with the journal’s requirements. For example, level 1 headings are now font size 18, level 2 headings are font size 16, all headings are written in sentence case, and the correct symbols to identify sets of equal contributors have been used. 

Author Response: We apologise for this mismatch. 

Author Action: We have now ensured that all grant information and numbers provided in ‘Funding Information’ match those included in the manuscript. 

"The study was supported by Kennedy Trust for Rheumatology Research (grant number: 171806), Versus Arthritis (grant number: 22473), Centre for OA Pathogenesis Versus Arthritis (grant numbers: 21621, 20205), Galapagos, Biosplice, Novartis, Fidia, UCB, Pfizer (non consortium member) and Somalogic (in kind contributions). " 

Author Response: Thank you for highlighting this omission. The funders had no role in any of these activities.

Author Action: We have added the suggested text to the cover letter and also added this statement at lines 637-643. 

"YD, TAP, PH, SL, AS, NKA, DF, BM, AMV, SK, VB, JMA and VK declare no conflicts of interest. FW has received consultancy fees from Pfizer, and has a leadership role at the Medical Research Council (panel member) and Osteoarthritis and Cartilage (Associate Editor). LSL has received consultancy fees from Arthro Therapeutics AB, and is an advisory board member of AstraZeneca (non consortium member). LJD has received consultancy fees from Nightingale Health PLC. TLV has no conflicts to declare with the exception of grant income for STEpUP OA from industry partners (see above). RAM is a shareholder of AstraZeneca. SB and JM are employees and shareholders of Novartis (consortium members). MK has received support for attending the Gordon Research Conference, OARSI meeting, International Cartilage Repair Society, Munster University, is a board member of the Dutch Arthritis Society (Chair of Visitation Board), and has a leadership role at Osteoarthritis Research Society International (Board of Directors Member). CTA has received consultancy fees from Novartis, and has received honoraria for educational purposes also from Novartis. DAW has received consultancy fees from GlaxoSmithKline plc, AKL Research & Development Limited, Pfizer Ltd, Eli Lilly and Company, Contura International, and AbbVie Inc, has received honoraria for educational purposes from Pfizer Ltd and AbbVie Inc, is a board member of UKRI (Director) and Versus Arthritis Advanced Pain Discovery Platform. "

Author Response: Thank you for highlighting this requirement. The competing interests declared do not in themselves affect data sharing. Data sharing is covered by a consortium agreement which all members signed. Its requirements are outlined in our response to point 5. 

Author Action: We have added the following statement to the Competing Interest Statement section as requested (lines 661-662): “This does not alter our adherence to PLOS ONE policies on sharing data and materials (see also Data Availability statement)”. However, we have also further clarified in this section the restrictions on sharing of data and/or materials as our obligations to the consortium to the response to point 5 below. We have added the updated competing interests section to our cover letter.

Author response: As part of legal restrictions (outlined in a consortium agreement, signed ahead of this work by all consortium members), and in line with the ethical approvals/ requirements of the contributing cohorts, the full STEpUP OA dataset cannot be made publicly available, either as files or in a public repository at any time. There are routes to access these primary datasets (discovery, replication) for bone fide research by researchers outside of the consortium after the publication of our primary analysis results. The minimal datasets upon which this data relies and all R code have been made publicly available, which allow the checking of our analysis and results whilst being in line with our consortium obligations, which we hope satisfies the journal and reviewers.

Author action: Our Data Availability Statement has been revised as follows (lines 718-724): “The minimal datasets upon which this data relies and all R code, including the html vignette, are available at https://github.com/dengyun-git/STEpUp_QC_Paper. The full STEpUP OA dataset may be made available by application to the Data Access and Publication Group of STEpUP OA (stepupoa@kennedy.ox.ac.uk) once the primary analysis manuscript is published, in accordance with what is stipulated in our Consortium Agreement. Neither the minimal dataset nor the full STEpUP OA dataset include patient identifiable data. This may attract a fee for access to cover administrative processing”. 

Author Response & action: Thank you for clarifying this. Both instances of “data not shown” have been removed (lines 351 & 356).

Author Response & action: This has been corrected (lines 947-1022).

8. We notice that your supplementary [Figure S1-S9] are included in the manuscript file. Please remove them and upload them with the file type 'Supporting Information'. Please ensure that each Supporting Information file has a legend listed in the manuscript after the references list.

Author Response & action: This has now been corrected.

Comments from PLOS Editorial Office:

1) We note that one of the reviewers expressed concerns about the quality of the images. Although the figures appear to be of poor quality when viewed as part of the pdf, the individual figure files are of a good quality. The original figures can be accessed by clicking on the link at the top of each page with a figure to download the file. Therefore, you do not need to produce new versions of the figures at this time, although please see note 8 above.

Author response: Thank you for clarifying this.

2) Although there are comments from 2 reviewers below, reviewer 2 felt unable to fully assess your submission. Please ignore the comments provided by reviewer 2.

Author response: As instructed, we have only responded to comments provided by reviewer 1.

Reviewer#1:

1. * Data availability- the data from the main study (proteomics) is not open access data. The mention of a fee to access needs more detail.

Author response: We apologise that our submitted manuscript did not have the minimal dataset and all R code, including the html vignette through the link provided. 

Author action: The data access statement has been amended as follows: “The minimal datasets upon which this data relies and all R code, including the html vignette, are available at https://github.com/dengyun-git/STEpUp_QC_Paper. The full STEpUP OA dataset may be made available by application to the Data Access and Publication Group of STEpUP OA (stepupoa@kennedy.ox.ac.uk) once the primary analysis manuscript is published, in accordance with what is stipulated in our Consortium Agreement. Neither the minimal dataset nor the full STEpUP OA dataset include patient identifiable data. This may attract a fee for access to cover administrative processing”. 

We can confirm that the Github link was activated ahead of submission of this revision.

2. * Abstract: The use of abbreviations should be removed in the abstract for easier reading. Indeed, some abbreviations (synovial fluid) are not even defined.

Author Response & action: Thank you. All abbreviations have been spelled out in full in the abstract. The abstract has also been edited to bring it down to within the allowed word count.

3. * Introduction: Minor: Check abbreviations are defined the first time used (eg FDA).

Author Response: Thank you. These have been checked and corrected (e.g. FDA).

4. Mixture of UK and US English, be consistent and use UK English.

Author Response: We note the need for consistency. 

Author Action: We believe we have identified and changed all instances, e.g. standardisation, normalisation, recognise etc to UK English. For instance, ‘recognised’ (line 4), ‘localised’ (line 5), ‘utilised’ (line 28) etc.

Methods

5. More details on the SF preparation from collection to storage are required. What was the time from collection to centrifugation, were samples subaliquot, where samples collected into low protein bind tubes? How long were samples stored at -80 for prior to analysis? Was the supernatant removed following centrifugation? Were samples kept in a fridge at all times? Was a cold centrifuged used? Were samples with blood contamination excluded? Were samples were collected following joint lavage what buffer was used and what were the protocols?

Author response: We apologise that in trying to reduce word count, insufficient detail was provided on our laboratory methods. In addition, some of this information was available in Supplementary Methods which we appreciate may have been overlooked. It should be noted that there would be slight methodological variations across the various contributing cohorts (as SF samples were already collected). We provide here the typical approach to SF preparation, including for the host institution cohorts and also for de novo collections as per our SOP for the sample processing. 

Answers to the reviewer’s specific questions are given below.

• What was the time from collection to centrifugation, were samples subaliquot, where samples collected into low protein bind tubes? 

- Typical time from collection to centrifugation was 2 hours. Samples were subaliquoted into 500 ul aliquots in 2 ml cryovials (not low bind).

• How long were samples stored at -80 for prior to analysis? 

- Time of storage (referred to as ‘sample age’) varied within and between cohorts and was recorded and used as a covariate in the Principal Components (PC) analysis.

• Was the supernatant removed following centrifugation? 

- Yes, for centrifuged samples only the supernatants were removed and stored down. As noted, some cohorts in replication had samples that were not centrifuged and these data were handled differently.

• Were samples kept in a fridge at all times? 

- No, the preference was to process as quickly as possible. In our experience, refrigerating SF makes it viscous and harder to centrifuge.

• Was a cold centrifuged used? 

- A temperature controlled centrifuge was used, typically set to 20 deg C, for reason above.

• Were samples with blood contamination excluded? 

- No they were not, but as listed (in tables S2 and S7), blood was graded and considered in our QC pipeline and as a covariate in the PC analysis.

• Were samples collected following joint lavage what buffer was used and what were the protocols? - Samples were never collected by joint lavage. All were collected directly from the joint, typically by needle arthrocentesis.

Author action: This additional information and other relevant points are now provided in Supplementary methods (p1-3).

6. Details of why certain criteria were used for filtering should be given from line 238-251.

Author response: Filters were agreed in principle in advance by our Data Analysis Group, based on typical approaches to proteomic data and are outlined in the initial QA plan (https://www.kennedy.ox.ac.uk/oacentre/step

---

## [Decision Letter · Decision Letter 1]

7 May 2024

PONE-D-23-35180R1Development of molecular endotype discovery from synovial fluid of individuals with knee osteoarthritis: the STEpUP OA ConsortiumPLOS ONE

Dear Dr. Perry,

Thank you for submitting your manuscript to PLOS ONE. Although the paper has been revised, residual concerns were expressed that should be addressed during the second round of revision. We invite you to submit a revised version of the manuscript after you have addressed all points by the reviewers.

We look forward to receiving your revised manuscript.

Kind regards,

Andre van Wijnen

Academic Editor

PLOS ONE

Reviewers' comments:

Reviewer's Responses to Questions

**Comments to the Author**

1. If the authors have adequately addressed your comments raised in a previous round of review and you feel that this manuscript is now acceptable for publication, you may indicate that here to bypass the “Comments to the Author” section, enter your conflict of interest statement in the “Confidential to Editor” section, and submit your "Accept" recommendation.

Reviewer #2: (No Response)

Reviewer #3: All comments have been addressed

2. Is the manuscript technically sound, and do the data support the conclusions?

Reviewer #2: (No Response)

Reviewer #3: Yes

3. Has the statistical analysis been performed appropriately and rigorously? 

Reviewer #2: (No Response)

Reviewer #3: Yes

4. Have the authors made all data underlying the findings in their manuscript fully available?

Reviewer #2: (No Response)

Reviewer #3: (No Response)

5. Is the manuscript presented in an intelligible fashion and written in standard English?

Reviewer #2: (No Response)

Reviewer #3: (No Response)

6. Review Comments to the Author

Reviewer #2: Apologies, but in the first review, I justified that I did not have sufficient knowledge to evaluate this article properly. Now, I have taken your review by mistake, as I act as a reviewer for many articles, and I realized it was the same article that I had previously declined to review. I'm sorry, but there is also no one from my research team available to conduct the review. I regret the inconvenience, but it would not be appropriate for me to assess the article.

Reviewer #3: Manuscript Number: PONE-D-23-35180R1

Article type: Research Article

Title of the paper: Development of molecular endotype discovery from synovial fluid of individuals with knee osteoarthritis: the STEpUP OA Consortium.

Short Title: Methods for molecular endotype discovery in STEpUP OA

Its nice to see that state-of-the-art technology is being used to analyze OA-SF involving a really large number of patients to unravel the disease pathology.

Following are my comments:

Comment 1.

It is an interesting thought that OA is actually not a single disease, but involves different types such as immune-mediated inflammation, mechanically-mediated inflammation (‘mechanoflammation’), low/failed tissue repair and cellular senescence may exist as distinguishable endotypes. However, all of these endotypes have significant proportion of overlapping pathophysiology which tend to blur the distinction between them. How the authors plan to address this complex issue?

Comment 2.

There is a lot of dynamics of protein variation in as per the grades of the OA (usually measured on KL grade scale) is this study is going to take the grades in account?

Comment 3:

If purpose of this communication is only to report a method/protocol for largescale analysis of synovial fluid proteins and not to report any of the endotypes, the paper title is misleading. Please find an appropriate title.

Comment 4:

If the authors are using ssDNA aptamers/SOMAmers, for harvesting proteins form synovial fluid. They must assure that the clinically obtained SF to be free from DNA and RNA. There is lot of DNA in synovial fluid (Ref. DOI: 10.1002/art.1780240905). Similarly, there are large number of micro and long non-coding RNAs present in the SF. How is the system free from their interreference?

Comment 5:

How the protocol reduce/deplete hugely abundant proteins like albumin, globulin form SF?

Comment 6

SF contain several proteases including very active tryptase and chymase, elastase etc. How have you prevented their action thereby loss of proteins on storage?

Comment 7

Why the Picture quality is so low?

7. PLOS authors have the option to publish the peer review history of their article (what does this mean?). If published, this will include your full peer review and any attached files.

Reviewer #2: No

Reviewer #3: **Yes: **Abhay Harsulkar

---

## [Author Response · Author response to Decision Letter 1]

7 Jun 2024

PLOS ONE Decision: Revision required [PONE-D-23-35180R1] - [EMID:8d32a3fbdc941e1f]

Manuscript Number: PONE-D-23-35180R1

Response to reviewer comments

We thank the reviewers and editorial team for their helpful comments, questions and suggestions and detail below a point-by-point response to these. Page numbers have been given with reference to the resubmitted tracked changed manuscript and supplementary materials.

Reviewer 3: 

It’s nice to see that state-of-the-art technology is being used to analyze OA-SF involving a really large number of patients to unravel the disease pathology.

Following are my comments:

1. Comment 1

It is an interesting thought that OA is actually not a single disease, but involves different types such as immune-mediated inflammation, mechanically-mediated inflammation (‘mechanoflammation’), low/failed tissue repair and cellular senescence may exist as distinguishable endotypes. However, all of these endotypes have significant proportion of overlapping pathophysiology which tend to blur the distinction between them. How the authors plan to address this complex issue?

Author response: Our aim in STEpUP OA was to establish whether there are distinct molecular endotypes discernible from synovial fluid (SF) proteins. Our methodology, which is shown in full in the next paper (now in preprint: https://www.medrxiv.org/content/10.1101/2024.06.05.24308485v1) (and can be reviewed in our analysis plan shown here: https://www.kennedy.ox.ac.uk/oacentre/stepup-oa) is to use unsupervised k-means clustering on the SomaScan data. However, to the authors comment, an inherent limitation of this type of analysis is that the analysis is biased towards finding endotypes if they have distinct rather than overlapping proteome profiles. Continuous protein profiles, or ones that are strongly overlapping, may still be clinically meaningful and this can be tested. We have mentioned in the discussion that we will use k means clustering of the data in the primary analysis of STEpUP OA. We feel that discussing the limitations of this is best left to the next manuscript as we are not discussing it further here.

Author Action: We have added additional text to help clarify (lines 558-560: “Our next step, the primary analysis of this dataset, seeks to answer definitively whether there are distinct discernible molecular endotypes, by unsupervised k means clustering, in this common, yet poorly understood disease”). 

2. Comment 2

There is a lot of dynamics of protein variation in as per the grades of the OA (usually measured on KL grade scale) is this study is going to take the grades in account?

Author Response: The primary focus of this manuscript was the development and validation of a methodology for large-scale analysis of synovial fluid proteins in knee OA which was inclusive of multiple radiographic grades. The grading approach employed for all cases was the KL grading system where it was available (in 766 OA cases). For the others we were able to categorise them into non-advanced (KL 0-2) and advanced radiographic OA (KL 3-4) by cohort eligibility criteria. A wide range of KL grades were intentionally included, to be as generalisable as possible. The association of differing radiographic grades with protein abundance is already included as part of our primary analysis plan (https://www.kennedy.ox.ac.uk/oacentre/stepup-oa); as above, the link to this plan is included at various points in this manuscript. A description of the cohorts and the variables included in the analysis, including their range in radiographic severity by KL grade can be found in S1 Table (final column). 

Author Action: We have added additional information on the radiographic scoring systems used in the methods with reference to S1 Table (lines 156-158: “Characteristics of cohorts and participant samples, including details of the radiographic scoring employed (Kellgren-Lawrence[58] grading where available), are described in S1 Table with core variable definitions also provided (S2 Table & Supplementary methods).”). 

We have added the following reference, cited as [58], to the text:

Kellgren, J.H. and J.S. Lawrence, Radiological assessment of osteo-arthrosis. Ann Rheum Dis, 1957. 16(4): p. 494-502.

3. Comment 3

If purpose of this communication is only to report a method/protocol for largescale analysis of synovial fluid proteins and not to report any of the endotypes, the paper title is misleading. Please find an appropriate title.

Author Response: Thank you. We do wish to make it clear that this is primarily a methodology paper and not the primary outcome of the analysis. 

Author Action: We have changed the title to: “Development of methodology to support molecular endotype discovery from synovial fluid of individuals with knee osteoarthritis: the STEpUP OA Consortium”

4. Comment 4

If the authors are using ssDNA aptamers/SOMAmers, for harvesting proteins form synovial fluid. They must assure that the clinically obtained SF to be free from DNA and RNA. There is lot of DNA in synovial fluid (Ref. DOI: 10.1002/art.1780240905). Similarly, there are large number of micro and long non-coding RNAs present in the SF. How is the system free from their interreference?

Author Response: We appreciate this comment. We believe that free DNA is unlikely to lead to significant assay interference as the SOMAmer technology relies on high affinity binding with specificity and selectivity for protein targets. Low affinity binding nucleic acids are removed during the assay process. As a result, any residual nucleic acids present in the SF are unlikely to interfere with the binding of SOMAmers to their protein targets.

Author Action: A comment regarding the presence of non-specific nucleic acids in the SF has now been added to the Discussion (lines 535-538: “Free nucleic acids in biological fluids such as SF could potentially interfere with the SomaScan assay. However, this is unlikely as the aptamers used for protein identification are designed to be high affinity for that protein with low affinity binders being competed off through application of an unlabeled polyanion competitor [72].”). 

Reference [72] has been added to the text:

Williams, S.A., et al., Plasma protein patterns as comprehensive indicators of health. Nat Med, 2019. 25(12): p. 1851-1857.

5. Comment 5

How the protocol reduce/deplete hugely abundant proteins like albumin, globulin form SF?

Author response: 

There are no steps for depletion of highly abundant proteins. One of the advantages of SomaLogic technology is that aptamer binding quantification can be carried in complex samples across a wide range of protein concentrations. This is achieved by dividing the sample into three dilutional ‘bins’ (1:5, 1:200 and 1:20,000) which enables high abundant proteins to be examined separately from low abundant ones. 

Author Action: We have added to Methods the following (lines 126-127: “Samples were assayed in three ‘dilution bins’ (to allow accurate quantification of high and low abundant proteins simultaneously) on the SomaLogic SomaScan Discovery Plex V4.1 by SomaLogic, in Boulder, US.)”). 

6. Comment 6

SF contain several proteases including very active tryptase and chymase, elastase etc. How have you prevented their action thereby loss of proteins on storage?

Author response: We are unable to control fully for such activities within the SF. To minimise these, we recommended strict adherence to standardised protocols for sample handling and storage to reduce protease activation and protein degradation. This included minimising freeze-thaw cycles (aliquoting individual samples prior to freezing), and ensuring consistent storage conditions to prevent fluctuations in temperature that may affect protease activity. Processing in the laboratory was also standardised to reduce enzymic activity at the time of analysis. Furthermore, we looked carefully for confounding technical factors that could have reflected poor protein stability (or active degradation) by including “age of sample”, “number of freeze-thaws”, “spun status” in our quality analysis planning. We report these findings in this manuscript for this reason and incorporated filters for proteins that were poorly stable on freeze thaw/storage as likely to be less informative.

Author Action: We have added the following text to the Methods: (lines 542-545: “…without a unified pre-specified sample processing protocol. This made distinguishing technical and biological variation, including through protease activity, difficult (as was the case for the intracellular protein score, which could reflect either variation in joint biology or variation in sampling handling).” 

7. Comment 7

Why the Picture quality is so low?

Author response & action: High quality images were submitted but likely compressed during first review phase. We will ensure that higher quality images are available in the final version.

---

## [Decision Letter · Decision Letter 2]

10 Jul 2024

PONE-D-23-35180R2Development of methodology to support molecular endotype discovery from synovial fluid of individuals with knee osteoarthritis: the STEpUP OA ConsortiumPLOS ONE

Dear Dr. Perry,

Thank you for submitting your manuscript to PLOS ONE. After careful consideration, we feel that it has merit but does not fully meet PLOS ONE’s publication criteria as it currently stands. Therefore, we invite you to submit a revised version of the manuscript that addresses the points raised during the review process.

The remaining issues are not major, but please answer the questions from a reviewer and an editor.

We look forward to receiving your revised manuscript.

Kind regards,

Masao Tanaka

Academic Editor

PLOS ONE

Journal Requirements:

Additional Editor Comments:

Reviewers responded that the revise generally addresses the issues.

However, reviewer 4 pointed out two remaining issues.

And Editor has a confirmation question.

On page P12, lines 193-195, authors refer to Cip as the logarithmic value of the protein concentration of sample i. Since the addition of the logarithm of a number is the logarithm of the multiplication of that number, isn't IPSi the logarithm after the addition?

That is, IPSi=logΣdpCip

Reviewers' comments:

Reviewer's Responses to Questions

**Comments to the Author**

1. If the authors have adequately addressed your comments raised in a previous round of review and you feel that this manuscript is now acceptable for publication, you may indicate that here to bypass the “Comments to the Author” section, enter your conflict of interest statement in the “Confidential to Editor” section, and submit your "Accept" recommendation.

Reviewer #3: (No Response)

Reviewer #4: All comments have been addressed

2. Is the manuscript technically sound, and do the data support the conclusions?

Reviewer #3: Yes

Reviewer #4: Yes

3. Has the statistical analysis been performed appropriately and rigorously? 

Reviewer #3: Yes

Reviewer #4: Yes

4. Have the authors made all data underlying the findings in their manuscript fully available?

Reviewer #3: Yes

Reviewer #4: Yes

5. Is the manuscript presented in an intelligible fashion and written in standard English?

Reviewer #3: Yes

Reviewer #4: Yes

6. Review Comments to the Author

Reviewer #3: Manuscript Number: PONE-D-23-35180R2

Article Type: Research Article

Full Title: Development of methodology to support molecular endotype discovery from synovial fluid of individuals with knee osteoarthritis: the STEpUP OA Consortium

The authors have satisfactorily addressed all the questions raised, expect following two points:

First:

Interreference of DNA form SF: the SF DNA can bind to the single stranded aptamers/somamers and therefore the related proteins may be underrepresented. Similarly, the micro and lnRNAs may also bind to the ssDNA aptamers and therefore may interfere the protein detection. The SF may be treated with DNAse and RNAse before proceeding.

Second:

SF protease may hydrolyse proteins upon long-term storage. The protease action has nothing to do with any freeze-thaw cycles or temperature fluctuations. I may suggest that the authors may use commercially available cocktail of proteinase inhibitors containing TPCK, TLCK, PMSF etc. This will effectively reduce protein degradation in long-term storage.

Reviewer #4: The authors appropriately responded to all of the comments the reviewer had raised. This manuscript now can be publishable.

7. PLOS authors have the option to publish the peer review history of their article (what does this mean?). If published, this will include your full peer review and any attached files.

Reviewer #3: **Yes: **Abhay Harsulkar

Reviewer #4: No

---

## [Author Response · Author response to Decision Letter 2]

22 Jul 2024

PLOS ONE Decision: PLOS ONE Decision: Revision required [PONE-D-23-35180R2]

Full Title: Development of methodology to support molecular endotype discovery from synovial fluid of individuals with knee osteoarthritis: the STEpUP OA Consortium

Response to reviewer comments

We thank the reviewers and editorial team for their helpful comments, questions and suggestions and detail below a point-by-point response to these. Page numbers have been given with reference to the resubmitted tracked changed manuscript and supplementary materials.

Editor:

Reviewers responded that the revise generally addresses the issues. However, reviewer 4 pointed out two remaining issues. And Editor has a confirmation question.

1) Comment 1

On page P12, lines 193-195, authors refer to Cip as the logarithmic value of the protein concentration of sample i. Since the addition of the logarithm of a number is the logarithm of the multiplication of that number, isn't IPSi the logarithm after the addition?

That is, IPSi=logΣdpCip

Author response:

We thank the editor for pointing this out. The equation is correct, but the text immediately before and after it missed out the word "log" in two places, which made it inaccurate. We have fixed this now. 

Author action: 

We have made the following two changes to the text: 

• line 191: "weighted sum of log protein concentrations"

• line 194: "difference in log concentration"

Reviewer 3:

The authors have satisfactorily addressed all the questions raised, expect following two points:

2) Comment 1

Interreference of DNA form SF: the SF DNA can bind to the single stranded aptamers/somamers and therefore the related proteins may be underrepresented. Similarly, the micro and lnRNAs may also bind to the ssDNA aptamers and therefore may interfere the protein detection. The SF may be treated with DNAse and RNAse before proceeding.

Author response:

We thank the reviewer for the comment. By incorporating DNase and RNase treatments, it may have been possible to reduce potential interference from DNA and RNA. However, our adopted protocol was in accordance with standard procedures using SomaScan (King et al. 2020; Rydén et al. 2024). It is worth noting that the binding affinity of aptamers is enhanced by chemical modification of the DNA.

• Rydén M, et al. Exploring the Early Molecular Pathogenesis of Osteoarthritis Using Differential Network Analysis of Human Synovial Fluid. Mol Cell Proteomics. 2024 Jun;23(6):100785. doi: 10.1016/j.mcpro.2024.100785. Epub 2024 May 14. PMID: 38750696.

• King JD et al. Joint Fluid Proteome after Anterior Cruciate Ligament Rupture Reflects an Acute Posttraumatic Inflammatory and Chondrodegenerative State. Cartilage. 2020 Jul;11(3):329-337. doi: 10.1177/1947603518790009. Epub 2018 Jul 22. PMID: 30033738; PMCID: PMC7298591.

Author action:

We have added the following text to the discussion (see lines: 535 to 540): “Free nucleic acids in biological fluids such as SF could potentially interfere with the SomaScan assay. However, this is unlikely as the aptamers used for protein identification, which are chemically modified DNA molecules, are high affinity for that protein allowing low affinity binders to be competed off through application of an unlabeled polyanion competitor[72]. It might have been possible to reduce interference further by pre-treating SF with DNAse or RNAse, but this is not considered standard for SomaScan analysis[73, 74].”. 

We have also added the above two references to the text (line 540). 

3) Comment 2

SF protease may hydrolyse proteins upon long-term storage. The protease action has nothing to do with any freeze-thaw cycles or temperature fluctuations. I may suggest that the authors may use commercially available cocktail of proteinase inhibitors containing TPCK, TLCK, PMSF etc. This will effectively reduce protein degradation in long-term storage.

Author response: We agree that this needs to be included as a comment for considering design of future studies although in this case all of our samples were pre-collected so this was not an option at the time of study development.

Author action:

We have added the following line to the discussion (see lines: 547 to 549): “Intrinsic protease activity within the fluids may have contributed to confounding associated with sample age. Addition of protease inhibitors at the time of collection could be something to consider in future sample collections, but is generally not the norm.”

Reviewer 4:

The authors appropriately responded to all of the comments the reviewer had raised. This manuscript now can be publishable.

---

## [Editor Report · Decision Letter 3]

16 Aug 2024

Development of methodology to support molecular endotype discovery from synovial fluid of individuals with knee osteoarthritis: the STEpUP OA Consortium

PONE-D-23-35180R3

Dear Dr. Perry,

We’re pleased to inform you that your manuscript has been judged scientifically suitable for publication and will be formally accepted for publication once it meets all outstanding technical requirements.

Kind regards,

Masao Tanaka

Academic Editor

PLOS ONE

Additional Editor Comments (optional):

With regard to the Editor's question, the definition of IPSi has bocome clear and not misleading, and as for the two questions raised by reviewer 3, they have been addressed in a realistic manner.

Therefore, all questions have been answered appropriately.

---

## [Editor Report · Acceptance letter]

3 Sep 2024

PONE-D-23-35180R3 

PLOS ONE

Dear Dr. Perry, 

I'm pleased to inform you that your manuscript has been deemed suitable for publication in PLOS ONE. Congratulations! Your manuscript is now being handed over to our production team.

Kind regards, 

on behalf of

Dr. Masao Tanaka 

Academic Editor

PLOS ONE